# Liver ChREBP deficiency inhibits fructose-induced insulin resistance in pregnant mice and female offspring

Jiaqi Li[1,5], Shuang Zhang [ID][2,5], Yuyao Sun[1], Jian Li[3], Zian Feng[4], Huaxin Li[2], Mengxue Zhang[2], Tengteng Yan[2], Jihong Han [ID][1✉] & Yajun Duan [ID][4✉]

## Abstract

**High fructose intake during pregnancy increases insulin resistance (IR) and gestational diabetes mellitus (GDM) risk. IR during pregnancy primarily results from elevated hormone levels. We aim to determine the role of liver carbohydrate response element binding protein (ChREBP) in insulin sensitivity and lipid metabolism in pregnant mice and their offspring. Pregnant C57BL/6J wild-type mice and hepatocyte-specific ChREBP-deficient mice were fed with a high-fructose diet (HFrD) or normal chow diet (NC) pre-delivery. We found that the combination of HFrD with pregnancy excessively activates hepatic ChREBP, stimulating progesterone synthesis by increasing MTTP expression, which exacerbates IR. Increased progesterone levels upregulated hepatic ChREBP via the progesterone-PPARγ axis. Placental progesterone activated the progesterone-ChREBP loop in female offspring, contributing to IR and lipid accumulation. In normal dietary conditions, hepatic ChREBP modestly affected progesterone production and influenced IR during pregnancy. Our findings reveal the role of hepatic ChREBP in regulating insulin sensitivity and lipid homeostasis in both pregnant mice consuming an HFrD and female offspring, and suggest it as a potential target for managing gestational metabolic disorders, including GDM.**

**Keywords** ChREBP; Progesterone; MTTP; Insulin Resistance; Pregnancy
**Subject Categories** Metabolism; Molecular Biology of Disease

## Introduction

Gestational diabetes mellitus (GDM) is a frequent complication characterized by insulin resistance (IR) during pregnancy (Buchanan and Xiang, 2005). Investigating IR during pregnancy is crucial in understanding the factors contributing to the development of GDM and enhancing the maternal and neonatal outcomes associated with this disorder. Excess dietary fructose intake has been considered a risk factor for pathological IR and GDM. There are human observational and controlled studies identifying that excessive sweetened food and beverage consumption causes poor pregnancy outcomes (Borgen et al, 2012; Englund-Ögge et al, 2012; Halldorsson et al, 2010). Fasting serum fructose is associated with risk of GDM (Zhang et al, 2022). Animal studies have provided evidence that maternal consumption of fructose during pregnancy leads to elevated maternal IR, hepatic steatosis, and increased susceptibility to GDM (Liu et al, 2022; Mukai et al, 2013; Zou et al, 2012). Meanwhile, fructose intake during pregnancy disrupts offspring lipid and glucose metabolic function, and the extent of this effect varies between the genders of offspring (Regnault et al, 2013; Vickers et al, 2011). However, the mechanism underlying these perturbations in dam and offspring is ill-defined, and the available remedies are currently limited.

Currently, it is widely believed that the sharp increase in hormone levels during pregnancy is the primary cause of IR and GDM (Kampmann et al, 2019). Progesterone is a cholesterol-derived hormone, which plays a critical role in mammalian reproduction (Kolatorova et al, 2022). Meanwhile, progesterone is thought to be one of the main factors of IR in GDM (Di Cianni et al, 2003). Progesterone induces apoptosis of beta cells and accelerates the development of diabetes in female *db/db* mice (Nunes et al, 2014; Picard et al, 2002). It was also noted that progesterone increases blood glucose levels via hepatic progesterone receptor membrane component 1 (Lee et al, 2020). In addition, progesterone aggravates IR by reducing the expression of glucose transporter 4 (GLUT4) (Sugaya et al, 2000).

ChREBP is a transcription factor that can be activated by fructose and its metabolites, and activated ChREBP can promote the expression of its downstream glycolytic and lipogenic genes (Kim et al, 2017; Sargsyan et al, 2023). However, the role of hepatic ChREBP in regulating systemic and hepatic glucose homeostasis and insulin sensitivity remains controversial. Studies have shown that liver-specific ChREBP KO mice had impaired insulin sensitivity under both normal diet and high-fat diet (Jois et al, 2017). Meanwhile, overexpression of a

[1]College of Life Sciences, State Key Laboratory of Medicinal Chemical Biology, Key Laboratory of Bioactive Materials of Ministry of Education, Nankai University, Tianjin, China. [2]Key Laboratory of Metabolism and Regulation for Major Diseases of Anhui Higher Education Institutes, College of Food and Biological Engineering, Hefei University of Technology, Hefei, China. [3]Tianjin Central Hospital of Gynecology Obstetrics, Tianjin 300052, China. [4]Department of Cardiology, The First Affiliated Hospital of USTC, Division of Life Sciences and Medicine, University of Science and Technology of China, Hefei, China. [5]These authors contributed equally: Jiaqi Li, Shuang Zhang. ✉E-mail: jihonghan2008@nankai.edu.cn; yajunduan@ustc.edu.cn

constitutively active form of ChREBP in the male mouse liver increased hepatic steatosis but enhanced insulin sensitivity under high-fat diet (Benhamed et al, 2012). What's more, hepatic ChREBP preserved glucose and lipid homeostasis under high-glucose diet in male mice (Sargsyan et al, 2023). On the contrary, inactivation of ChREBP in *ob/ob* male mice improved glucose tolerance (Dentin et al, 2006). These results remind us that the effects of hepatic ChREBP on insulin sensitivity and glucose homeostasis may be related to specific physiological or pathological states or nutritional contexts (Katz et al, 2021). Based on the background, we aim to determine the specific role of hepatic ChREBP in HFrD-induced IR and lipid metabolic disorder during pregnancy, in which the metabolic states change largely and frequently.

# Results

## The changes in insulin sensitivity and lipid metabolism are associated with hepatic ChREBP activation during normal pregnancy

Normal pregnancy leads to benign and transitory IR from mid-to-late gestation in order to meet the energy demands of the pregnant dam and fetus (Liu et al, 2020; McIntyre et al, 2019). We collected serum and liver samples from C57BL/6J mice in different stages of pregnancy (Fig. 1A). We observed an increase in serum insulin and glucose levels as well as HOMA-IR index (Appendix Fig. S1A–C) and a decrease in insulin signaling pathway sensitivity from mid to late gestation (Appendix Fig. S1D). Firstly, we examined the mRNA expression of ChREBP in highly expressed tissues and found that the most pronounced increase in ChREBP was observed in the liver and lasted throughout the development of pregnancy (Fig. 1B). Subsequently, we assessed ChREBP protein expression in tissues with high levels of expression and observed the most significant elevation was in the liver, which persisted throughout the entire pregnancy (Fig. 1C,D; and Appendix Fig. S1E). The expression of PKLR and SCD1 also increased and lasted throughout pregnancy (Appendix Fig. S2A). Consistently, the levels of serum and hepatic lipids rose continuously (Appendix Fig. S2B–G). Taken together, the changes in insulin sensitivity and lipid accumulation may be related to the upregulation of hepatic ChREBP during normal pregnancy.

## IR and hepatic steatosis exacerbated by an HFrD during pregnancy are associated with the overactivation of hepatic ChREBP

Next, we constructed a model of severe IR during pregnancy by feeding pregnant mice with an HFrD (Fig. 1E). The serum insulin level, GTT and ITT assays showed that the consumption of HFrD during pregnancy (PF group) led to much more severe IR compared with mice that are pregnant (PC group) only or exposed to fructose only (CF group) (Fig. EV1A–C). Meanwhile, pregnancy plus HFrD significantly reduced insulin signaling pathway sensitivity (Fig. EV1D). Moreover, pregnancy plus HFrD feeding (PF group) caused small and irregular islets (Fig. EV1E), indicating that HFrD feeding increased the dysfunction of islets in pregnant mice. Compared with CF or PC groups, the islets with a smaller insulin-positive cell core surrounded by a mantle of alpha cells in PF group (Fig. EV1F). The RNA-seq analysis demonstrated that hepatic *Chrebp* and the expression of its

target gene PKLR and SCD1 was significantly increased in the mice with HFrD feeding during pregnancy, compared with NC feeding (PF vs. PC, Fig. 1F). The bubble chart of KEGG and GO pathway analysis showed that genes involved in fructose and lipid metabolism-related signaling pathways were significantly enriched (PF vs. PC, Fig. 1F). Meanwhile, compared with CF and PC groups, late pregnancy plus HFrD feeding increased the mRNA and protein levels of ChREBP in liver (PF vs. CF, PC, Figs. 1G,H and EV1G). HFrD feeding upregulated the expression of ChREBP-targeted proteins involved in lipogenesis, contributing to the development of hepatic steatosis (PF vs. CF, PC, Figs. 1H and EV1H,I).

In addition, we collected blood samples from pregnant people with or without GDM. *ChREBP* mRNA levels in the whole blood were increased in the GDM group compared with the control (Ctrl) group (Fig. 1I). Taken together, the results showed that HFrD feeding aggravates IR and hepatic steatosis associated with increased hepatic ChREBP in pregnancy.

## Hepatic ChREBP deficiency improves HFrD-induced IR and hepatic steatosis during pregnancy

To further explore the specific role of hepatic ChREBP in insulin sensitivity during pregnancy in vivo, we designed hepatocyte-specific ChREBP knockout (h*Chrebp* KO) mice and the control (*Chrebp*^flox/flox^) mice (Appendix Fig. S3A) and conducted the experiments as indicated in Fig. 2A. The HOMA-IR index and ITT assessments indicated that under normal dietary conditions during pregnancy, IR is modestly improved in hChREBP-KO mice (KPC vs. WPC). Under conditions induced by HFrD, IR is significantly improved in hChREBP-KO mice (KPF vs. WPF, Fig. 2B–E). GTT assay showed that HFrD-induced glucose intolerance was significantly improved in h*Chrebp* KO mice (KPF vs. WPF). However, under normal dietary conditions, the absence of ChREBP has no significant effect on glucose tolerance in mice (KPC vs. WPC, Fig. 2F). At the molecular level, ChREBP deficiency in hepatocytes improves the decreased sensitivity of the insulin pathway induced by HFrD (Appendix Fig. S3B). In addition, hepatocyte-specific ChREBP deficiency ameliorated the irregular pancreatic islet morphology and smaller insulin-positive cell core induced by HFrD (KPF vs. WPF, Fig. 2G,H). These results indicated that hepatocyte ChREBP deficiency significantly ameliorated HFrD-induced IR during pregnancy, but its effect on IR in mice undergoing normal dietary conditions during pregnancy is modest.

Meanwhile, we observed that the protein and mRNA expression of hepatic ChREBP and its downstream lipogenic genes was significantly decreased in h*Chrebp* KO mice and cannot be induced by HFrD feeding (KPF vs. KPC, Fig. 2I and Appendix Fig. S3C–E). Consistently, hepatocyte-specific ChREBP deficiency reduced HFrD-induced excessive lipid accumulation during pregnancy (KPF vs. WPF, Appendix Fig. S3F–J). Taken together, the results showed that hepatic ChREBP deficiency improves HFrD-induced IR and hepatic steatosis during pregnancy.

## Hepatic ChREBP forms a positive feedback loop with progesterone and contributes to HFrD-induced gestational IR and hepatic steatosis

Our findings revealed that hepatocyte-specific ChREBP deficiency ameliorates HFrD-induced IR during pregnancy. In the stage of pregnancy, the increased IR is caused by the insulin-desensitizing

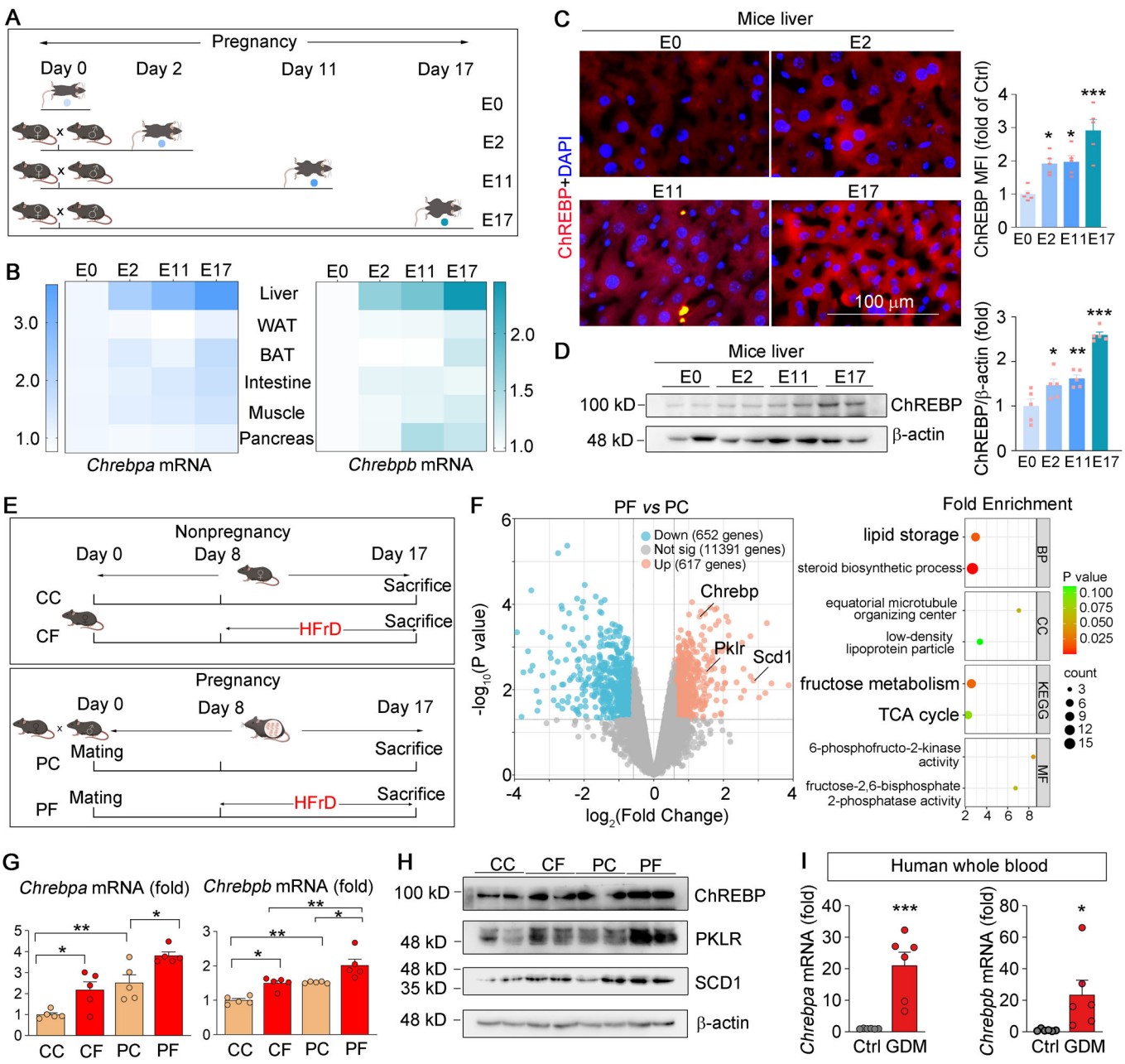

**Figure 1. Hepatic ChREBP expression is related to gestational IR.**

(A–D) Female C57BL/6J wild-type mice fed with NC were randomly divided into four groups. The E0 group was unmated and defined as Day 0 of embryonic development. The other three groups were mated with adult male mice overnight, and pregnant mice sacrificed on embryonic Day 2, Day 11, and Day 17 were called the E2, E11, and E17 groups (A); heatmap revealing *Chrebpa* and *Chrebpb* mRNA levels in mouse tissues at different stages of pregnancy (B); the protein levels of ChREBP in mouse liver were determined by immunofluorescent staining with quantification of the mean immunofluorescence intensity (MFI) of images (C); mouse liver samples were collected and the protein levels of ChREBP were determined by Western blot with quantification of the band density (D), $n = 5$. (E–H) Female C57BL/6 J wild-type mice were divided into four groups: CC group, C57BL/6J mice that received NC; CF group, C57BL/6 J mice that received HFrD from Day 8 to Day 17 of pregnancy; PC group, pregnant mice that received NC; PF group, pregnant mice that received HFrD from Day 8 to Day 17 of pregnancy (E); RNA-seq assay with mouse liver total RNA in PF and PC groups. Volcano plot showing the transcripts in liver that were decreased (blue dots), increased (orange dots), or not changed (gray dots), $n = 3$ (F); expression of *Chrebpa* and *Chrebpb* mRNA was determined by qRT-PCR, $n = 5$ (G); expression of ChREBP and its downstream target protein was determined by Western blot in liver (H). (I) Whole blood was obtained from control pregnancy women and GDM patients. RNA was extracted by the blood total RNA kit (Zomanbio, China), and *Chrebpa* and *Chrebpb* mRNA levels were evaluated by qRT-PCR, $n = 6$. Data information: All graphs are represented as Mean ± SEM, $n$: biological replicates. In (C, D), One-way ANOVA followed by Tukey's multiple comparisons test was used. *$P < 0.05$, **$P < 0.01$, ***$P < 0.001$ vs. E0 group. In (F), GO and KEGG enrichment analysis was performed with the R package clusterProfiler, with a Bonferroni correction and an adjusted $p$-value of 0.05. In (G), One-way ANOVA followed by Tukey's multiple comparisons test was used. *$P < 0.05$, **$P < 0.01$ vs. the indicated group. In (I), *$P < 0.05$, ***$P < 0.001$ vs. Ctrl group. Source data are available online for this figure.

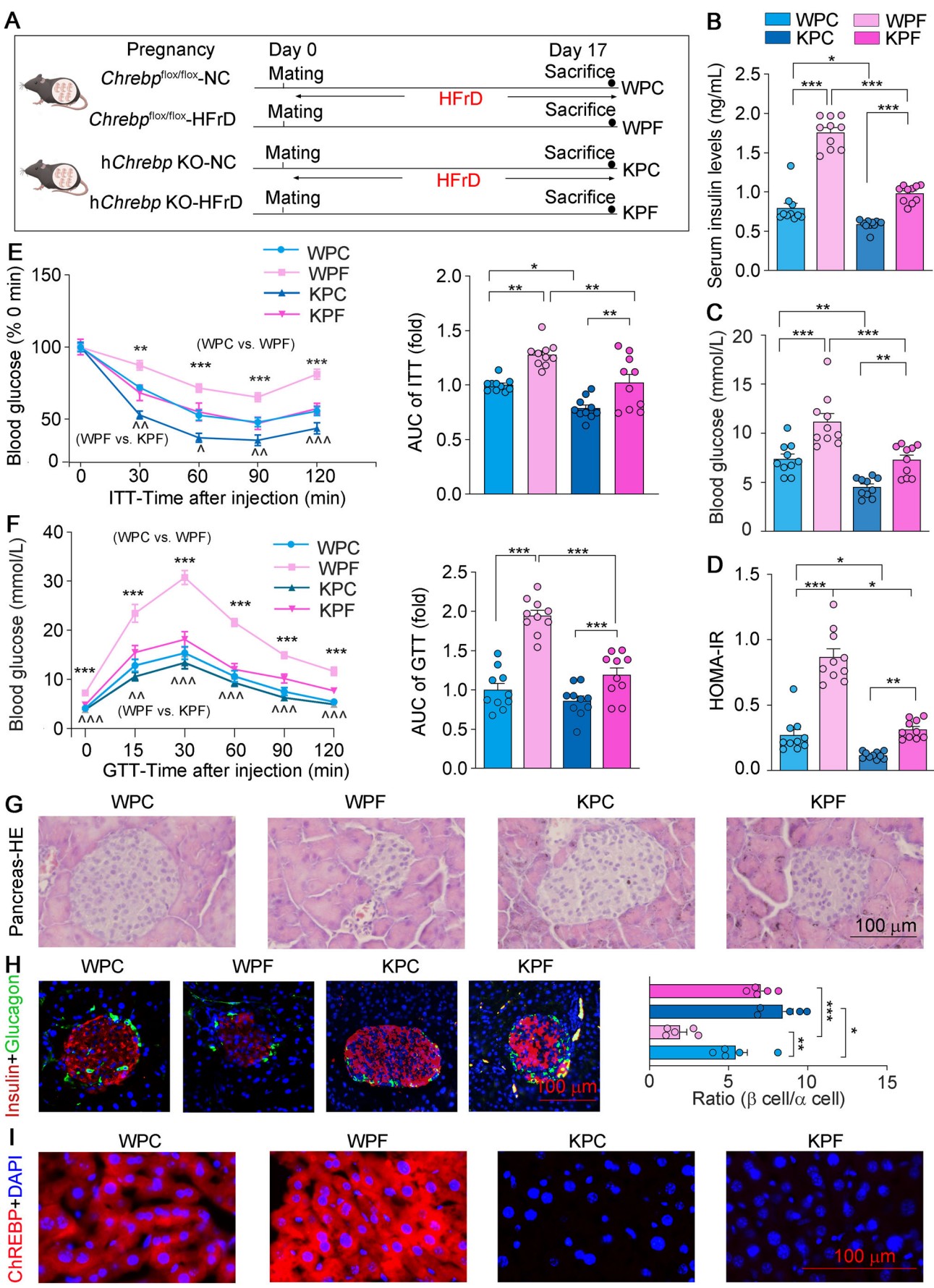

**Figure 2. Hepatic ChREBP deficiency improves HFrD-induced IR.**

(A) Female *Chrebp*<sup>flox/flox</sup> and hepatocyte-specific *Chrebp* KO (h*Chrebp* KO) mice were mated with adult male mice overnight on Day 0 and then divided into four groups that received NC or HFrD from Day 0 to Day 17: WPC (*Chrebp*<sup>flox/flox</sup> mice fed NC), WPF (*Chrebp*<sup>flox/flox</sup> mice fed HFrD), KPC (h*Chrebp* KO mice fed NC) and KPF (h*Chrebp* KO mice fed HFrD), all mice were sacrificed on Day 17. (B–D) Determination of serum insulin (B), blood glucose (C) levels, and HOMA-IR (D), $n = 10$. HOMA-IR was calculated as HOMA-IR = (Fasting blood glucose (mM) × Insulin (ng/mL))/22.5. (E, F) ITT (E) or GTT (F) assay at the E14 or E15, and quantitation of areas under curves (AUC), $n = 10$. (G, H) HE staining (G) and immunofluorescent staining (H) with anti-glucagon (green) or anti-insulin (red) antibodies of mice pancreas sections, and the ratio of beta cell area (insulin-positive) to alpha cell area (glucagon positive) was calculated, $n = 5$. (I) Expression of the ChREBP protein in the liver was determined by immunofluorescent staining. Data information: All graphs are represented as Mean ± SEM, $n$: biological replicates. Two-way ANOVA followed by Tukey's multiple comparisons test was used. In (B–D), $*P < 0.05$, $**P < 0.01$, $***P < 0.001$ vs. indicated group. In (E, F), $*P < 0.05$, $**P < 0.01$, $***P < 0.001$, WPC vs. WPF group, $^P < 0.01$, $^^P < 0.001$, WPF vs. KPF group. In (H), $*P < 0.05$, $**P < 0.01$, $***P < 0.001$ vs. indicated group. Source data are available online for this figure.

effects of gestational hormones (Buchanan and Xiang, 2005). Therefore, these results remind us to determine the relationship between hepatic ChREBP and pregnancy hormones. We found that ChREBP deficiency did not affect serum prolactin and PGLF (placental growth factor) levels (Fig. 3A,B). However, ChREBP deficiency inhibited the elevation of serum progesterone and its downstream metabolite estradiol levels induced by HFrD (KPF vs. WPF). Meanwhile, hepatocyte ChREBP deficiency had only a modest effect on serum progesterone and did not affect its downstream metabolite estradiol levels in pregnant mice fed a normal diet (KPC vs. WPC, Fig. 3C,D). We also found maternal serum progesterone levels were positively correlated with hepatic *Chrebp* mRNA expression in pregnant mice (Fig. 3E). Consistently, we collected serum (for serum progesterone levels determination) and whole blood (for whole blood *Chrebp* mRNA determination) from healthy (Ctrl) and GDM patients and found serum progesterone levels increased in GDM patients (Fig. 3F). Moreover, serum progesterone levels of pregnant people were also positively correlated with *ChREBP* mRNA levels in whole blood samples (Fig. 3G). These results indicate that the relationship between hepatic ChREBP and serum progesterone may be involved in regulating HFrD-induced IR in pregnancy.

To further confirm the relationship between ChREBP and progesterone levels, we next conducted the experiments both in vitro and in vivo. First, we determined that progesterone-induced ChREBP expression at mRNA and protein levels in both a concentration- and time-dependent manner in mice primary hepatocytes (Appendix Fig. S4A,B). However, estradiol did not affect ChREBP expression at either the transcriptional or translational level (Appendix Fig. S4C,D), indicating that progesterone, but not estradiol, may be involved in the regulation of hepatic ChREBP levels during pregnancy. To further demonstrate that progesterone can directly modulate hepatic ChREBP expression in vivo, we intraperitoneally administered female C57BL/6J mice with saline solution (CTRL) or progesterone (PRO) for continuous 12 days (Fig. 3H). Progesterone injection increased serum progesterone levels significantly (Fig. 3I). The expression of hepatic ChREBP and its target genes were induced by progesterone at the same time (Fig. 3J,K). Consistently, progesterone injection also increased hepatic TG and FFA contents (Fig. 3L–N). In conclusion, a positive feedback mechanism between ChREBP and progesterone may be an important factor for regulating insulin sensitivity and lipid accumulation during pregnancy under metabolic disorders.

## Hepatic ChREBP deficiency decreases progesterone levels by inhibiting MTTP in liver

Subsequently, we investigated the specific mechanism by which hepatic ChREBP regulates progesterone levels. It is worth noting

that progesterone production during pregnancy primarily occurs in the placenta and ovaries but not in the liver (Filipovich et al, 2016). Our findings revealed that progesterone levels in the placenta and ovaries were reduced in h*Chrebp* KO mice compared with *Chrebp*<sup>flox/flox</sup> mice (Appendix Fig. S5A,B). However, hepatic ChREBP deficiency did not affect the expression of genes involved in progesterone synthesis in the ovary and placenta (Appendix Fig. S5C,D). Next, we performed RNA-seq analysis of mouse liver in KPF and WPF groups. The results of GO/KEGG enrichment analysis indicated cholesterol metabolism and PPAR pathways were involved in the most significantly enriched signaling pathways (Fig. 4A). We then determined that hepatocyte *Chrebp* deficiency decreased cholesterol levels of circulation, placenta, and ovaries in HFrD-fed pregnant mice, while hepatic cholesterol levels remained unaffected (KPF vs. WPF, Fig. 4B,C; Appendix Fig. S5E,F). These findings implied that hepatic ChREBP deficiency may have an impact on the hepatic elimination of cholesterol into the bloodstream. The results of RNA-seq assay demonstrated that microsomal triglyceride transfer protein (*Mttp*), but not other cholesterol metabolism-related genes (*Apob*, *Abca1*, *Abcg1*) expression was significantly decreased in the h*Chrebp*-KO pregnancy dam with HFrD, compared with the *Chrebp*<sup>flox/flox</sup> pregnancy dam with HFrD feeding (KPF vs. WPF, Fig. 4D). Consistently, we probed these genes expression and observed that hepatic ChREBP deficiency reduced the mRNA and protein expression of MTTP (Fig. 4E; Appendix Fig. S5G), which has been previously demonstrated to be the ChREBP target gene (Niwa et al, 2018).

To provide additional evidence for the involvement of MTTP in the regulation of progesterone levels by hepatic ChREBP, we overexpressed hepatic MTTP by an adeno-associated virus (AAV) vector under a constitutively active, liver-specific promoter (TBG) (Fig. 4F). The results indicated that overexpression of hepatic MTTP in hepatocyte-specific *Chrebp*-deficient mice decreased the level of hepatic cholesterol level, but led to elevated levels of cholesterol and progesterone in circulation (Appendix Fig. S6; Fig. 4G–I). Furthermore, the results of HOMA-IR, ITT, and GTT assays demonstrated that the beneficial effect of hepatocyte ChREBP deficiency on IR and glucose intolerance was blocked by hepatic MTTP overexpression (Fig. 4J–N). Therefore, hepatocyte-specific *Chrebp* deficiency inhibited IR during pregnancy by inhibiting the MTTP-progesterone axis (Fig. 4O).

## Progesterone induces hepatic ChREBP expression in a PPARγ-dependent manner

To investigate the underlying mechanisms by which progesterone-induced ChREBP expression, we determined that progesterone activated *ChREBP* promoter activity (Appendix Fig. S7A).

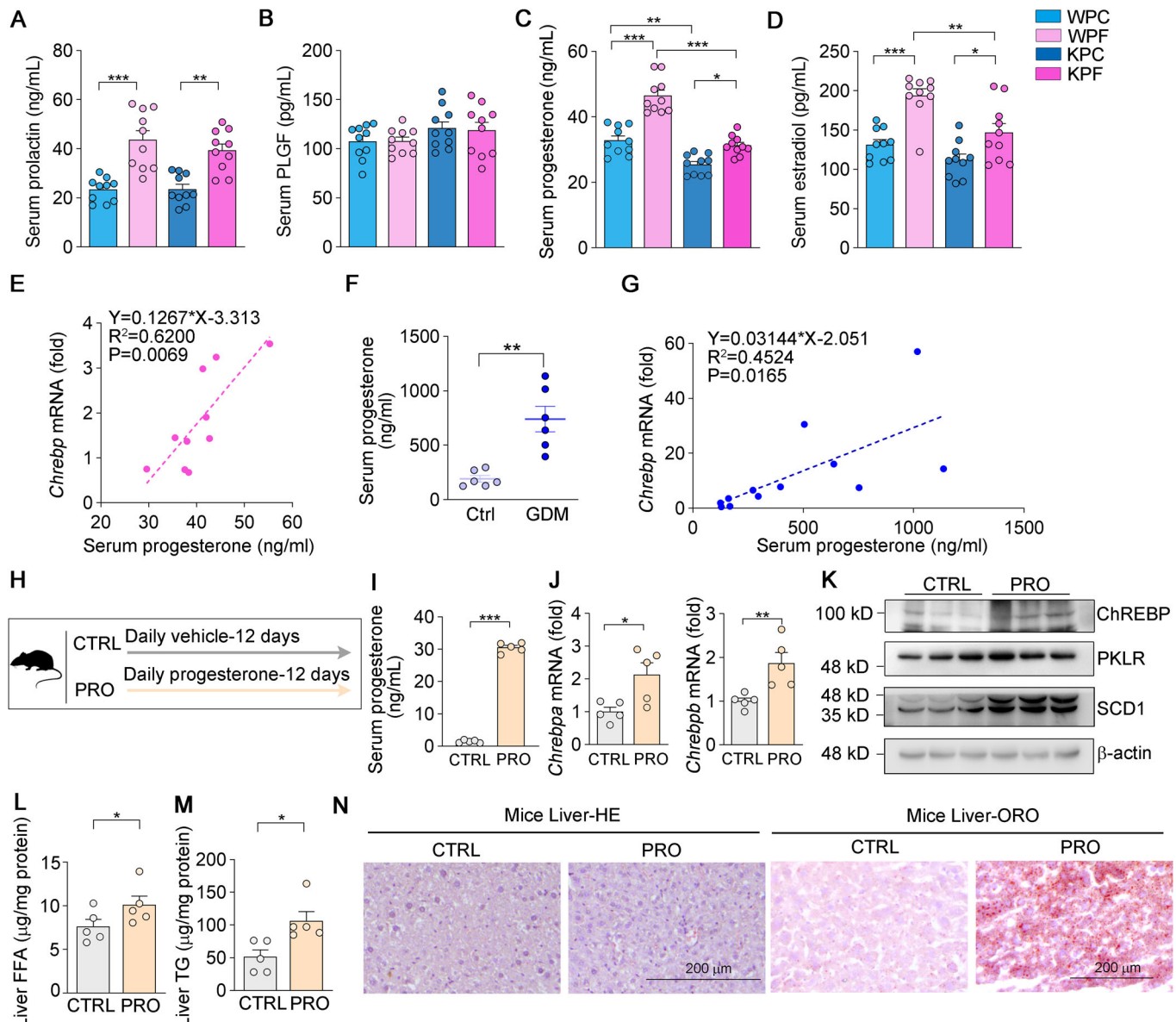

**Figure 3. Hepatic ChREBP expression is positively correlated with progesterone levels.**

(A–D) Determination of serum prolactin levels (A), PLGF levels (B), progesterone levels (C), and estradiol levels (D) of mice, $n = 10$. (E) The correlations between serum progesterone and hepatic *Chrebp* levels in WPC and WPF group, $n = 5$. (F, G) Serum progesterone levels in control and GDM patients were determined (F) and the correlation between serum progesterone and blood *ChREBP* mRNA levels were analyzed (G), $n = 6$. (H–N) Female C57BL/6J wild-type mice were intraperitoneally injected with vehicle (saline solution) or progesterone (1 mg/mouse) dissolved in saline solution for 12 consecutive days and the mice were defined as CTRL or PRO (H); determination of serum progesterone levels (I); expression of *Chrebpa* and *Chrebpb* mRNA was determined by qRT-PCR in mouse liver (J); expression of ChREBP and its downstream target protein was determined by Western blot in mouse liver (K); mouse liver was conducted free fatty acid (FFA) (L) and triglyceride (TG) (M) quantitative analysis with total liver lipid extract; HE staining and Oil Red O (ORO) staining (N), $n = 5$. Data information: All graphs are represented as Mean ± SEM, $n$: biological replicates. $*P < 0.05$, $**P < 0.01$, $***P < 0.001$ vs. the indicated group. (A–D): Two-way ANOVA followed by Tukey's multiple comparisons test was used. (F, I, J, L, M): Two-tailed t-tests were performed to assess differences between two experimental groups. Source data are available online for this figure.

Therefore, we treated primary hepatocytes with progesterone and mifepristone (a progesterone receptor antagonist) and determined that progesterone-increased ChREBP expression is dependent on progesterone receptor (PGR) (Appendix Fig. S7B). To evaluate the binding effects of PGR to ChREBP promoter regions, we searched for putative Progesterone Response Element (PRE) motifs in the promoter of *ChREBP*. However, we did not detect PRE in ChREBP.

Considering that our previous research has found that progesterone increased PPARγ expression in a PGR-dependent manner and the binding of PGR with the progesterone response element in the PPARγ promoter (Yang et al, 2016), and the KEGG/GO analysis indicated that the PPAR pathway was enriched in hepatic ChREBP deficiency mice (Fig. 4A), we proposed that progesterone might indirectly induce ChREBP expression through the activation of

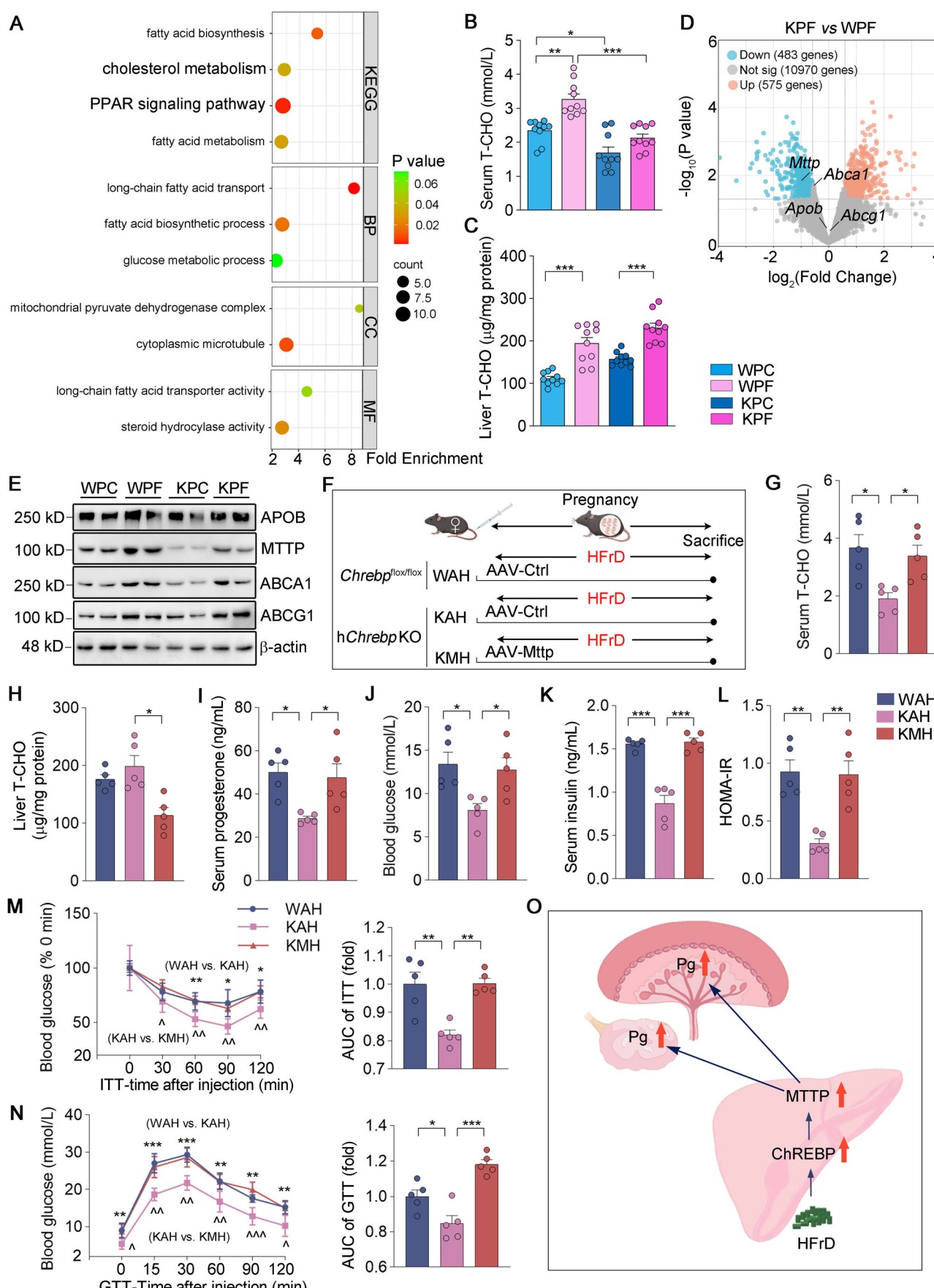

**Figure 4. Hepatic ChREBP deficiency inhibits progesterone synthesis by decreasing MTTP expression in liver.**

(A) RNA-seq assay with mouse liver total RNA in KPF and WPF groups. Enrichment analysis of GO and KEGG pathway, $n = 3$. (B, C) Determination of total cholesterol (T-CHO) levels in serum (B) and liver (C) of mice, $n = 10$. (D) RNA-seq assay with mouse liver total RNA in KPF and WPF groups. Volcano plot showing the transcripts in liver that were decreased (blue dots), increased (orange dots), or not changed (gray dots), $n = 3$. (E) Expression of protein involved in cholesterol transport was determined by Western blot in mouse liver. (F–O) Female *Chrebp*flox/flox and h*Chrebp* KO mice were administered AAV-Ctrl or AAV-Mttp through the tail vein. Subsequently, they were mated with adult male mice overnight and subjected to an HFrD from Day 0 to Day 17 (F); determination of total cholesterol (T-CHO) levels in serum (G); determination of total cholesterol (T-CHO) levels in liver (H); determination of progesterone levels in serum (I); determination of blood glucose levels (J); serum insulin (K); HOMA-IR (L); HOMA-IR was calculated as HOMA-IR = (Fasting blood glucose (mM) × Insulin (ng/mL))/22.5; ITT (M) or GTT (N) assay at the E14 or E15, and quantitation of areas under curves (AUC). $n = 5$. (O) A diagram illustrating the mechanism of progesterone activation by hepatic ChREBP. Pg, progesterone. Data information: All graphs are represented as Mean ± SEM, $n$: biological replicates. In (A), GO and KEGG enrichment analysis was performed with the R package clusterProfiler, with a Bonferroni correction and an adjusted $p$-value of 0.05. In (B, C), Two-way ANOVA followed by Tukey's multiple comparisons test was used. *$P < 0.05$, **$P < 0.01$, ***$P < 0.001$ vs. the indicated group. In (D), Two-tailed t-tests were performed to assess differences between two experimental groups. $P < 0.05$. In (G–N), Two-way ANOVA followed by Tukey's multiple comparisons test was used. *$P < 0.05$, **$P < 0.01$, ***$P < 0.001$ vs. the indicated group. Source data are available online for this figure.

PPARγ. To further determine whether progesterone-induced hepatic ChREBP expression is related to PPARγ, we mutated a putative PPARγ responsive element (*PPRE*) in the proximal region ($+14 \sim +26$) of the human *ChREBP* promoter (Fig. 5A) and determined that the normal *ChREBP* promoter rather than the mutated *ChREBP* promoter was activated by progesterone and PPARγ-specific agonist rosiglitazone (Fig. 5B), indicating that the putative *PPRE* plays an important role in PPARγ-activated *ChREBP* transcription. The results of the ChIP assay further confirmed the binding of PPARγ with *PPRE* in the *ChREBP* promoter and the binding capacity was further enhanced by the activation of PPARγ (Fig. 5C). In vivo, the activation of PPARγ by rosiglitazone increased ChREBP expression in mouse liver (Appendix Fig. S7C,D).

To determine whether progesterone-induced ChREBP expression is related to PPARγ in vivo, we constructed hepatocyte-specific *Pparg* knockout (h*Pparg* KO) mice and littermate control (*Pparg*flox/flox) mice. The basal level of ChREBP was decreased in the liver of h*Pparg* KO mice (Fig. 5D,E). Firstly, female *Pparg*flox/flox and h*Pparg* KO mice were mated with males, and liver samples were collected on pregnancy day 17 (Fig. 5F). Pregnancy upregulated the expression of hepatic ChREBP in *Pparg*flox/flox mice but not in h*Pparg* KO mice (Fig. 5G,H). To further rule out the interference by other hormones, female *Pparg*flox/flox and h*Pparg* KO mice were directly injected with progesterone for 12 days (Fig. 5I). *Pparg* deficiency inhibited the increase in progesterone levels (Fig. 5J). Progesterone injection upregulated the expression of ChREBP and its downstream lipogenic genes in *Pparg*flox/flox mice but not in h*Pparg* KO mice (Fig. 5K,L), indicating that progesterone-induced ChREBP expression in a PPARγ-dependent manner. Consistently, hepatocyte-specific *Pparg* deficiency inhibited hepatic steatosis caused by progesterone (Appendix Fig. S7E–H). Taken together, these results suggested that ChREBP is a target of PPARγ in the liver and that progesterone-induced hepatic ChREBP is dependent on PPARγ.

## HFrD during pregnancy promotes metabolism disorder and activates ChREBP-progesterone loop in female offspring mice

As a placenta hormone, progesterone can enter into the circulation of offspring (Siemienowicz et al, 2020; Villee, 1969). Hence, an abundance of progesterone can potentially trigger the progesterone-ChREBP loop in the offspring through the placenta, subsequently influencing IR and hepatic steatosis in the offspring. To verify this conjecture, we conducted the same experiment as that in Fig. 1E, and the offspring were retained as shown in Fig. 6A.

Interestingly, IR was increased in female but not male offspring in the fPF group (Fig. 6B–E; Appendix Fig. S8A,B). Maternal HFrD feeding during pregnancy strongly increased serum progesterone levels and hepatic ChREBP mRNA levels in female offspring but had little effect on male offspring (Fig. 6F,G). Meanwhile, hepatic ChREBP levels showed a similar tendency to serum progesterone levels of female offspring, indicating the positive correlation between them (Fig. 6H). Moreover, maternal HFrD feeding during pregnancy upregulated hepatic ChREBP and lipid synthesis-related protein expression in female offspring (Fig. 6I), accompanied by increased serum and hepatic lipid levels (Fig. 6J–M; Appendix Fig. S8C). In conclusion, our results demonstrated that maternal HFrD intake during gestation can cause activation of the ChREBP-progesterone-ChREBP loop in female offspring, thereby promoting IR and hepatic steatosis.

## Maternal hepatic ChREBP plays a central role in HFrD-induced metabolic disorders in female offspring

To confirm that the selective damage in female offspring is dependent on the maternal ChREBP-progesterone loop, we conducted the same experiment as that in Fig. 2A and the offspring were maintained as shown in Fig. 7A. Maternal hepatocyte ChREBP deficiency improved IR in female offspring induced by HFrD during pregnancy (fKPF vs. fKPC, Fig. 7B–D). Moreover, maternal HFrD during pregnancy induced smaller insulin-positive cell core in female offspring but not male offspring, and maternal hepatic ChREBP deficiency rescued the impairment of pancreas (Appendix Fig. S9). In addition, insulin signaling was not inhibited in the fKPF-WT or fKPF-KO group (fKPF vs. fKPC, Fig. 7E). Maternal hepatic ChREBP deficiency had no significant effect on insulin sensitivity in male offspring (Fig. EV2A–D). These results indicated that the HFrD-activated-maternal ChREBP-progesterone loop can induce IR in female offspring caused by HFrD consumption during pregnancy.

Subsequently, we determined that maternal hepatic ChREBP deficiency inhibited the HFrD-induced ChREBP expression in female offspring (fKPF-WT vs. fKPC-WT, Fig. 7F,G). Furthermore, maternal hepatic ChREBP deficiency also leads to the suppression of ChREBP target genes and lipid accumulation in female offspring (fKPF-WT vs. fKPC-WT, Fig. 7H–L). Maternal hepatic ChREBP deficiency had no significant effect on ChREBP expression and lipid metabolism in male offspring (fWPF vs. fWPC, fKPF vs. fKPC, Fig. EV2E–K). These results indicated that maternal ChREBP-progesterone loop induced female offspring progesterone-ChREBP loop and then promoted IR and lipid metabolism disorder (Fig. 7M).

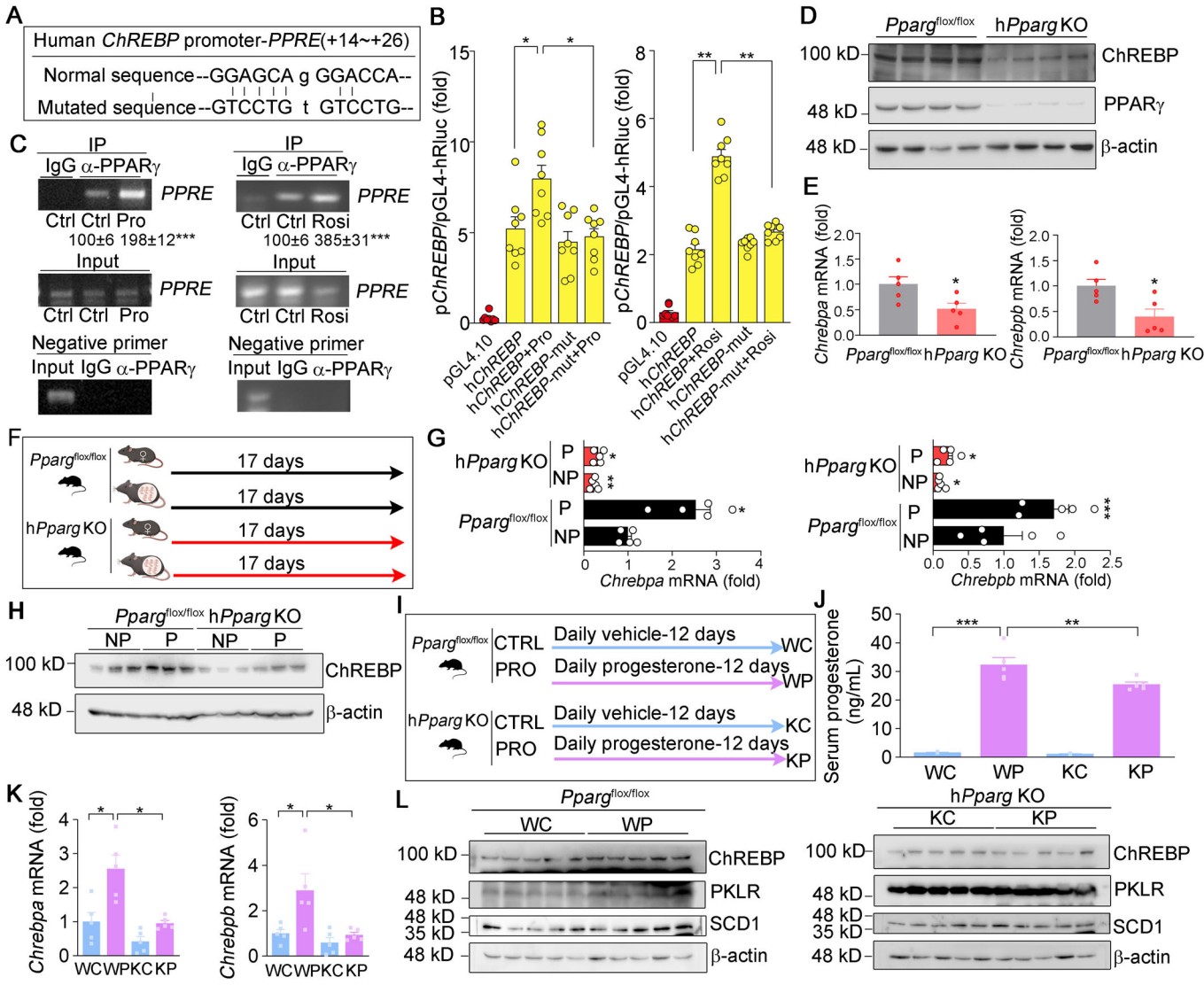

**Figure 5. Progesterone activates hepatic ChREBP expression through PPARγ.**

(A) Putative *PPRE* region in the h*ChREBP* promoter and the mutated sites. (B) HepG2 cells were transfected with the *ChREBP* promoter or *ChREBP-PPRE*-mut promoter and Renilla (as an internal control) overnight, followed by 100 nM progesterone or 10 µM rosiglitazone treatment for another 12 h. The activity of firefly and Renilla luciferases in the cellular lysate was determined by dual-luciferase reporter assay, $n = 8$. (C) HepG2 cells were treated with vehicle, 100 nM progesterone or 10 µM rosiglitazone for 16 h. Chromatin was then isolated from cells after sonication, followed by immunoprecipitation (IP) with normal IgG or anti-PPARγ antibodies. PCR was conducted with primers for the corresponding *PPRE* in the h*ChREBP* promoter. (D, E) Female *Pparg*flox/flox mice and h*Pparg* KO mice fed normal chow and liver tissues were subjected to Western blot (D) and qRT-PCR (E), $n = 5$. (F–H) Female *Pparg*flox/flox mice and h*Pparg* KO mice were mated with males, and liver samples were collected at pregnancy day 17 (F); expression of *Chrebpa* and *Chrebpb* mRNA was determined by qRT-PCR in mouse liver (G), $n = 5$; protein expression of ChREBP was determined by Western blot in mouse liver (H); NP, non-pregnancy; P, pregnancy. (I–L) Female *Pparg*flox/flox mice and h*Pparg* KO mice were randomly divided into four groups: WC and KC were *Pparg*flox/flox or h*Pparg* KO mice intraperitoneally injected with vehicle (saline solution); WP and KP were *Pparg*flox/flox or h*Pparg* KO mice intraperitoneally injected with progesterone solution (1 mg/day/mouse) for 12 consecutive days (I); determination of serum progesterone levels (J). (K, L) Expression of *Chrebpa* and *Chrebpb* mRNA was determined by qRT-PCR in mouse liver (K); expression of ChREBP and its downstream target protein was determined by Western blot in mouse liver (L); $n = 5$. Data information: All graphs are represented as Mean ± SEM, $n$: biological replicates. (B, C, G, J, K): Two-way ANOVA followed by Tukey's multiple comparisons test was used. (E): Two-tailed t-tests were performed to assess differences between two experimental groups. In (B), *$P < 0.05$, **$P < 0.01$ vs. the indicated group. In (C), ***$P < 0.001$ vs. the Ctrl-αPPARγ group. In (E), *$P < 0.05$ vs. the *Pparg*flox/flox group. In (G), *$P < 0.05$, **$P < 0.01$, ***$P < 0.001$ vs. the NP group. In (J, K), *$P < 0.05$, **$P < 0.01$, ***$P < 0.001$ vs. the indicated group. Source data are available online for this figure.

## Discussion

ChREBP represents promising targets for the development of therapies for the treatment of T2DM (Régnier et al, 2023), but its role in gestational diabetes has not been studied. Prior studies have shown that the effects of hepatic ChREBP on insulin sensitivity and glucose homeostasis may be related to specific physiological or pathological states or nutritional contexts (Katz et al, 2021). Our study demonstrated for the first time that hepatocyte-specific ChREBP deficiency ameliorates IR by inhibiting progesterone

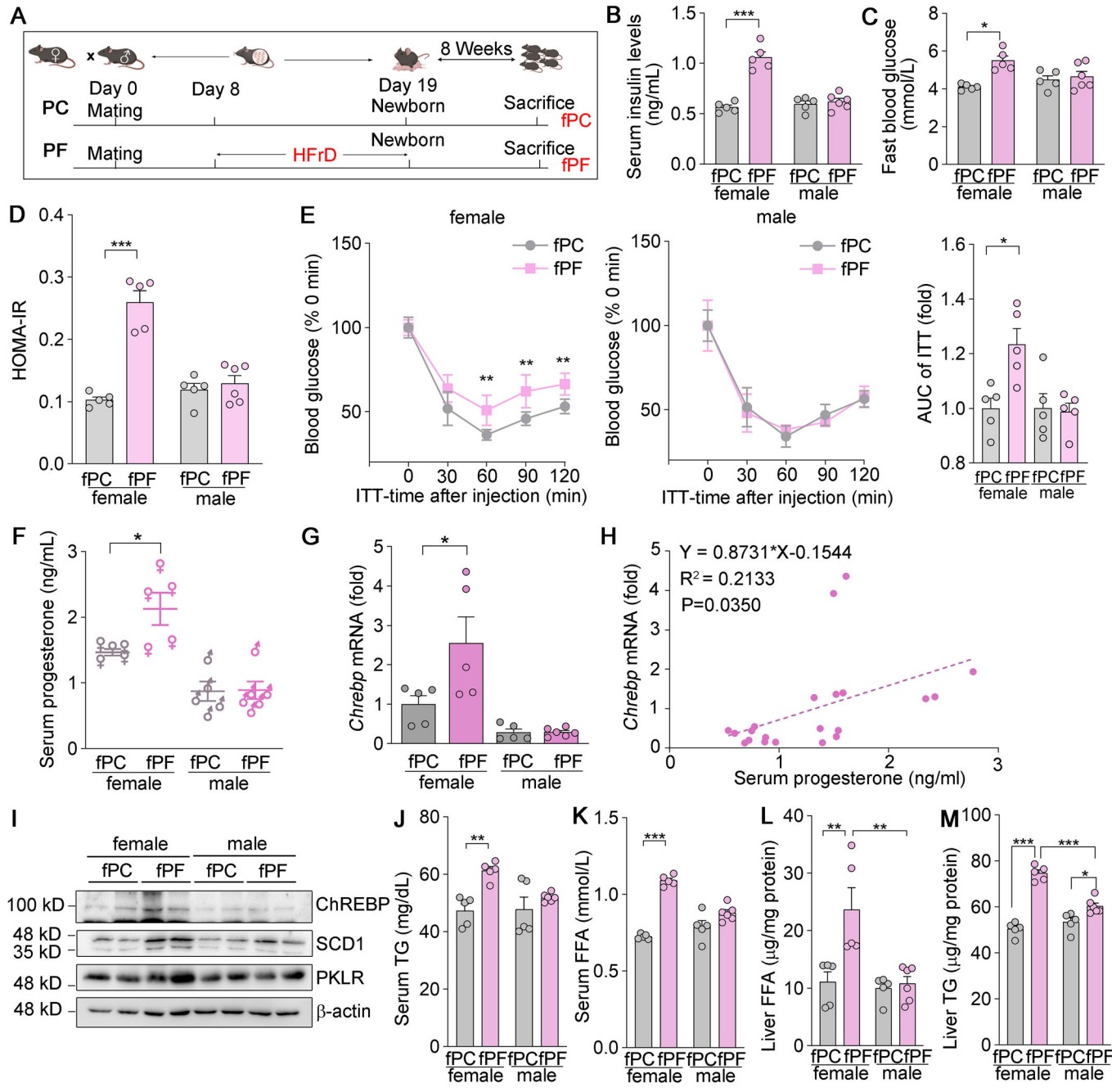

**Figure 6. HFrD during pregnancy promotes IR and activates ChREBP-progesterone loop in female offspring.**

(A) Offspring of the pregnant mice in the PC or PF groups (similar experiment to Fig. 1E) were, respectively, defined as the fPC or fPF group and maintained until 8 weeks old (sexual maturity). (B–D) Serum insulin levels (B), fasting blood glucose (C), and HOMA-IR (D) of offspring. (E) ITT assay was conducted, and the area under the curve (AUC) was calculated based on the ITT assay. (F–H) Serum progesterone levels (F) and hepatic *Chrebp* mRNA expression (G) in offspring (fPC, fPF) were determined, and the correlations between serum progesterone and hepatic *Chrebp* levels in offspring (H) were analyzed. (I) Expression of ChREBP and its downstream target protein was determined by Western blot in mouse liver. (J, K) Serum from offspring was subjected to TG (J) and FFA (K) quantitative analysis. (L, M) Livers from offspring were subjected to FFA (L) and TG (M) quantitative analysis, $n = 5–6$. Data information: All graphs are represented as Mean ± SEM, $n$: biological replicates. Two-way ANOVA followed by Tukey's multiple comparisons test was used. *$P < 0.05$, **$P < 0.01$, ***$P < 0.001$ vs. the indicated group. Source data are available online for this figure.

synthesis under high fructose conditions during pregnancy. Normally, progesterone levels remain relatively low in typical physiological conditions but experience a substantial increase throughout pregnancy (Sherer et al, 2017). Under high-fructose

stimulation, hepatic ChREBP expression increases and then elevates the levels of progesterone. Thus, the activation of the ChREBP-progesterone loop predominantly occurs during the distinctive physiological process of pregnancy under high-fructose

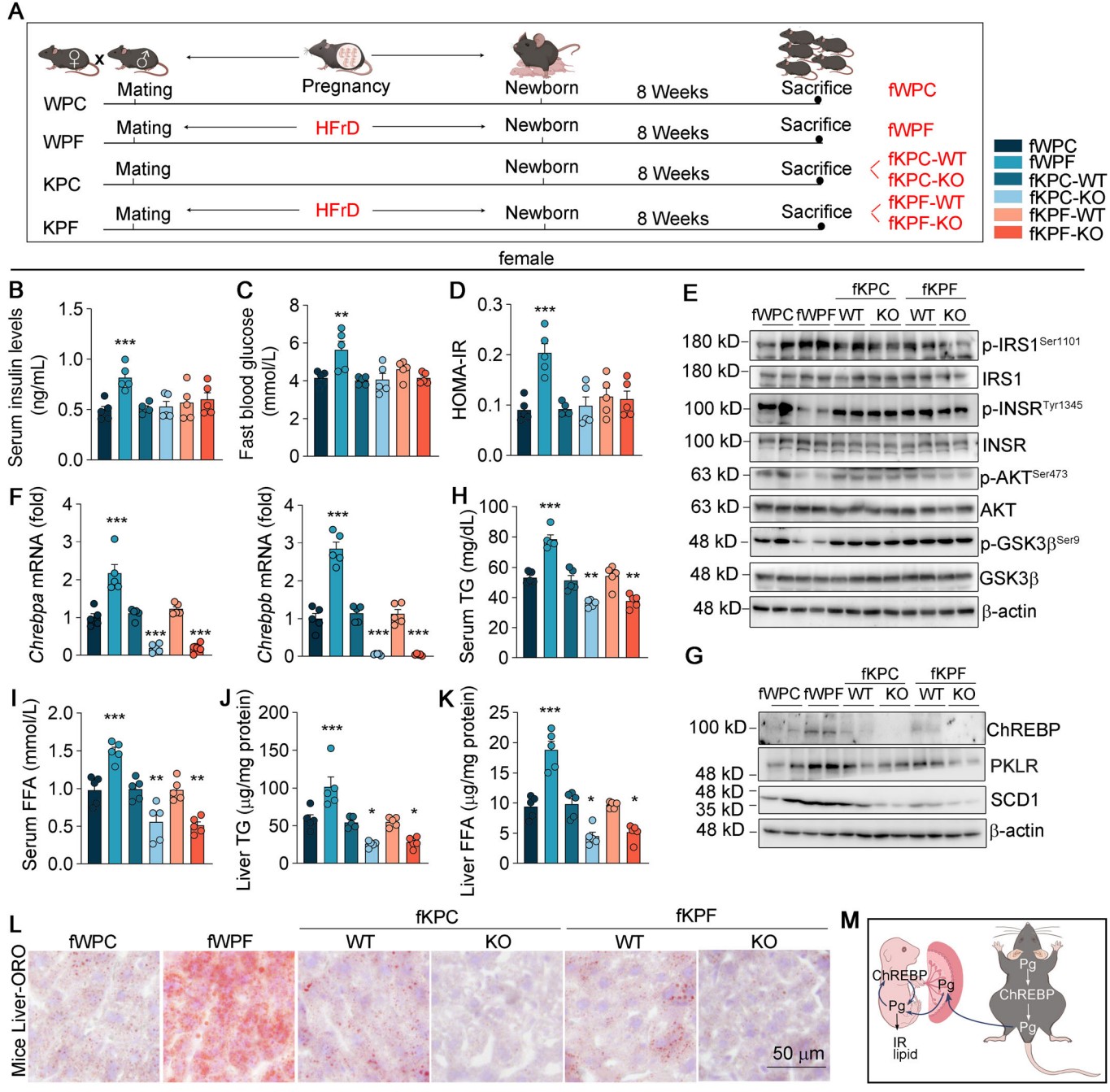

**Figure 7. Maternal hepatic ChREBP deficiency improves HFrD-impaired glucose and lipid homeostasis in female offspring.**

(A) Offspring from the pregnant mice in the WPC, WPF, KPC, and KPF groups (similar experiment to Fig. 2A) were, respectively, defined as the fWPC, fWPF, fKPC, and fKPF groups and maintained until 8 weeks of age (sexual maturity). (B–D) Serum insulin levels (B), fasting blood glucose (C), HOMA-IR index (D) of female offspring. $n = 5$. (E) Expression of protein related to insulin signaling pathway was determined by Western blot in the liver from female offspring. (F) Expression of *Chrebpa* and *Chrebpb* mRNA in the liver from female offspring was determined by qRT-PCR, $n = 5$. (G) Expression of ChREBP and its downstream target protein was determined by Western blot in the liver from female offspring. (H–K) Serum (H and I) and liver (J and K) from female offspring were subjected to triglyceride (TG) and free fatty acid (FFA) quantitative analysis. (L) The liver from female offspring was subjected to Oil Red O (ORO) staining. $n = 5$. (M) A diagram illustrating the mechanism of offspring progesterone-ChREBP axis activation by maternal. Pg, progesterone. Data information: All graphs are represented as Mean ± SEM, $n$: biological replicates. Owo-way ANOVA followed by Tukey's multiple comparisons test were used. *$P < 0.05$, **$P < 0.01$, ***$P < 0.001$ vs. fWPC group. Source data are available online for this figure.

feeding. In addition, we found that a high level of progesterone has no effect but overexpression of ChREBP promotes insulin sensitivity in vitro, respectively (Appendix Fig. S10), providing evidence that the effects of ChREBP-progesterone loop on insulin sensitivity are present during pregnancy under physiological conditions. We also observed that maternal hepatic ChREBP deficiency reduced the number of offspring (Appendix Table S2), which further supports the inhibitory effect of ChREBP on progesterone and the important role of hepatic ChREBP in energy supplements in pregnancy.

ChREBP exists in two isoforms, alpha and beta. ChREBPα is the primary isoform and is predominantly expressed in the liver. ChREBPβ is found in a wider range of tissues, including adipose tissue, muscle, and pancreas. It was noted that glucose-mediated activation of ChREBPα induces the expression of ChREBPβ that is transcribed from an alternative promoter. Therefore, the expression of ChREBPβ typically serves as a reliable indicator of tissue ChREBP activity (Herman et al, 2012). Our results demonstrate that pregnancy significantly increases ChREBP protein levels, with a further elevation in ChREBP protein expression observed during pregnancy with an HFrD. However, considering that detecting ChREBPβ protein levels in liver is not straightforward through Western blot analysis, we separately examined the changes in ChREBPα and ChREBPβ at the mRNA levels. The results revealed that pregnancy augments ChREBPβ at the mRNA level, and HFrD feeding during pregnancy further enhances the transcriptional expression of ChREBPβ. This provides compelling evidence for the activation of ChREBP in the liver during pregnancy. Meanwhile, we have observed that the downstream target genes of ChREBP (PKLR and SCD1) synchronously increase with hepatic ChREBP during pregnancy and under high-fructose conditions. However, it is worth noting that PKLR, SCD1 and other lipogenic molecules are also regulated by other transcription factors, and we are not yet clear about the specific mechanisms involved (Wu et al, 2017; Zhang et al, 2010). Given the multitude of molecules involved in regulating lipid synthesis during pregnancy, we speculate that other transcription factors or post-transcriptional regulatory mechanisms may control the expression of these molecules. The specific mechanisms still need to be validated through subsequent mechanistic experiments. This may require further extraction of hepatic nuclear proteins and multi-omics testing.

Progesterone is essential for normal uterine and oviduct function during pregnancy (Kolatorova et al, 2022), therefore, progesterone is often widely used as a fetal preservation drug in early pregnancy (Gao et al, 2022). However, the classic view held that the antagonistic effect of progesterone on insulin secretion during pregnancy is one of the main causes of GDM (Kühl, 1998). The onset of GDM typically occurs in the second trimester of pregnancy, when progesterone levels are high (Kim et al, 2002). Progesterone can cause decreased insulin sensitivity in multiple organs and tissues. Mechanistically, progesterone enhances IR through a reduction in GLUT4 expression (Campbell and Febbraio, 2002; Sugaya et al, 2000). In the pancreas, progesterone can be toxic to pancreatic beta cells through an oxidative stress-dependent mechanism to induce apoptosis (Nunes et al, 2014). In the liver, progesterone increases hepatic glucose production via PGRMC1, which may exacerbate hyperglycemia in diabetes (Lee et al, 2020). Our study determined that progesterone increases hepatic ChREBP expression through a positive feedback loop and is involved in

HFrD-induced gestational IR. Meanwhile, the activation of hepatic ChREBP by progesterone can cause hepatic lipid accumulation. Although the causal relationship between tissue lipid accumulation and IR is unclear (Beaven et al, 2013), lipid accumulation in hepatocytes has been associated with hepatic IR (Qu et al, 2019). Therefore, the lipid accumulation induced by ChREBP may be another reason for the promotion of progesterone to IR. These results prompted a need for caution when using progesterone analogs and managing high levels of progesterone during pregnancy.

It has been reported that maternal fructose intake during pregnancy leads to sex-specific changes in offspring endocrine function, but the extent of the effects on male and female offspring is still a subject of debate. Vickers et al found that maternal fructose intake during pregnancy and lactation increases blood glucose levels in female fetuses (Vickers et al, 2011). Moreover, Rodríguez et al found that liquid fructose intake in pregnancy exacerbates dyslipidemia in adult female offspring (Rodríguez et al, 2016a). However, another study has described that fructose (10% wt/vol) exposure during pregnancy leads to hyperinsulinemia, impaired insulin signaling, and low adiponectin levels in male but not female offspring (Rodríguez et al, 2016b). Furthermore, no studies have revealed the mechanism underlying the predilection for sex selection. Our results supported that HFrD-induced IR during pregnancy selectively impaired glucose and lipid metabolism and insulin sensitivity in female but not male offspring. We further indicated that maternal hepatocyte ChREBP deficiency ameliorated the disturbance of glucose and lipid metabolism and insulin sensitivity in female offspring caused by HFrD feeding during pregnancy, which is related to progesterone levels. Considering that male progesterone levels are inherently lower than those in females, and the expression of progesterone receptors in males is much lower than in females, the ChREBP-progesterone axis is not sensitive in males (Asavasupreechar et al, 2020).

Our primary focus in this study is on ChREBP levels in the liver. However, obtaining liver samples from pregnant individuals can be challenging and is often not feasible. Considering the challenge of obtaining liver samples during pregnancy, we aimed to explore whether ChREBP levels in more easily accessible blood samples could serve as a surrogate marker for ChREBP activity during pregnancy. We searched GEO database and found that the transcriptional level of ChREBP was significantly increased in the peripheral blood mononuclear cells of donors at the 9th month (late stage of pregnancy) of pregnancy compared with nonpregnant subjects (Appendix Fig. S11A). In mice blood, we also found that the transcriptional level of ChREBP was significantly increased in donors at the late stage of pregnancy compared with nonpregnant mice (Appendix Fig. S11B). Meanwhile, ChREBP mRNA levels in liver increase during mice pregnancy (Fig. 1B), suggesting that whole-blood ChREBP may serve as an indicator of the enhanced ChREBP expression in the liver during pregnancy. To further support this concept, compared with healthy pregnant women, ChREBP mRNA levels also increased in human GDM patients (Fig. 1I). In mouse blood samples, we also observed the increase of ChREBP mRNA level after HFrD-diet (Appendix Fig. S11C). Meanwhile, ChREBP mRNA levels in liver increase during mice pregnancy with HFrD (Fig. 1H). These findings demonstrated that blood ChREBP levels have consistency with hepatic ChREBP expression. Therefore, the mRNA levels of ChREBP in blood

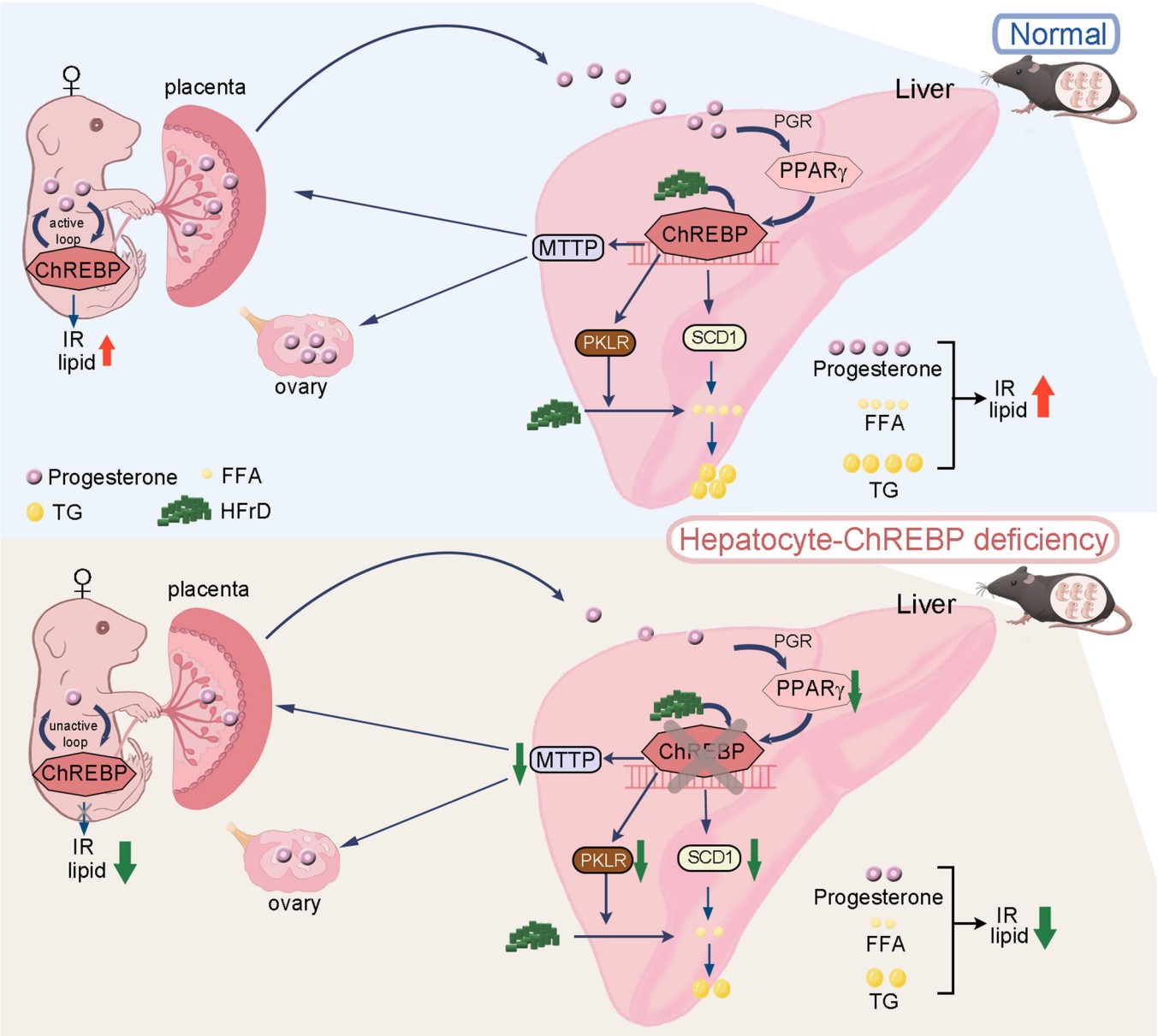

**Figure 8. Schematic representation for hepatic ChREBP deficiency-inhibited gestational IR and hepatic steatosis.**

In the case of HFrD feeding, hepatic ChREBP is doubly activated by fructose and pregnancy. Heptic ChREBP promoted MTTP expression and further increased progesterone levels in circulation. As progesterone levels increase, hepatic ChREBP expression is increased through a PPARγ dependent manner. The excessive activation of the progesterone-ChREBP axis results in severe lipid ectopic accumulation in the liver and systematic IR. Excess progesterone enters the offspring via the placenta, thereby activating the progesterone-ChREBP axis, resulting in IR and increased hepatic steatosis in the female offspring. When hepatic ChREBP is deficient, on one hand, the expression of downstream lipogenesis-related molecules is inhibited, which ameliorates hepatic steatosis, especially under high carbohydrate diets; on the other hand, the circulating progesterone level decreases, which can contribute to the improvement of HFrD-induced IR during pregnancy.

samples may serve as an alternative marker for hepatic ChREBP activity during pregnancy.

The limitations are listed as follows. Although our data indicated that overexpression of MTTP in hepatic ChREBP KO mice does increase circulating progesterone, the specific mechanism remains unclear. Considering that blood-derived cholesterol is a critical substrate for progesterone synthesis and it has been established that MTTP can regulate the efflux of cholesterol from the liver to the blood (Alam et al, 2001; Niwa et al, 2018; Tardif et al, 2014), we speculate that MTTP may promote progesterone synthesis by regulating cholesterol metabolism. However, increasing circulating cholesterol does not necessarily lead to an increase in steroid hormone synthesis. For example, statin drugs play a significant role in lowering circulating cholesterol in the body but do not affect circulating testosterone in men or estradiol in women (Oluleye et al, 2019). This may be because the amount of circulating

cholesterol far exceeds the quantity required for steroid hormone synthesis by many orders of magnitude. Therefore, MTTP may also potentially operate through indirect mechanisms mediated by lipid droplets or other effects on the liver, which warrants further investigation. Meanwhile, the experiment does not conclusively demonstrate that MTTP activation is the sole pathway through which ChREBP regulates progesterone in the liver, but it does suggest that MTTP may be an important mechanism for ChREBP-mediated regulation of progesterone levels. Considering these limitations, future research may involve liver-specific overexpression of ChREBP in mice to investigate its relationship with insulin resistance. Furthermore, more advanced studies could entail the injection of si-MTTP into liver-specific ChREBP-overexpressing mice to explore the role of MTTP in liver ChREBP and IR. In addition, isotopic labeling of cholesterol within the liver may be necessary to directly assess the impact of liver ChREBP knockout on the conversion of cholesterol into progesterone. We remain committed to ongoing research and further investigations to address these limitations.

In conclusion, our study links hepatic ChREBP to progesterone and HFrD-induced IR both in the pregnant dam and offspring female individuals for the first time and provides a new positive loop involved in this process. Briefly, in normal dietary conditions, hepatic ChREBP has only a modest effect on progesterone production and the IR of pregnancy. In the case of high fructose intake, ChREBP is over-activated, which leads to hyper-activation of progesterone-PPARγ-ChREBP-MTTP-progesterone loop and results in severe gestational IR and lipid ectopic accumulation. Moreover, excess progesterone enters the offspring via the placenta with the effects of activating ChREBP, resulting in IR and increasing hepatic steatosis in the female offspring (Fig. 8). Our findings unveiled the role of hepatic ChREBP in metabolic disorders caused by high fructose-containing foods and drinks during pregnancy, especially GDM. The precise maintenance of ChREBP-progesterone loop is essential in regulating lipid accumulation and insulin sensitivity in pregnant mice and future generations.

## Methods

### Reagents and tools table

| Reagent/Resource | Reference or Source | Identifier or Catalog Number |
|---|---|---|
| **Experimental Models** *List cell lines, model organism strains, patient samples, isolated cell types etc. Indicate the species when appropriate.* | | |
| Chrebp^flox/flox mice | GemPharmatech, Nanjing, China | RRID: IMSR_GPT: T009211 |
| HepG2 cell line | ATCC, USA | RRID: CVCL_0027 Cat# HB-8065 |
| **Recombinant DNA** *Indicate species for genes and proteins when appropriate* | | |

| Reagent/Resource | Reference or Source | Identifier or Catalog Number |
|---|---|---|
| **Antibodies** *Include the name of the antibody, the company (or lab) who supplied the antibody, the catalog or clone number, the host species in which the antibody was raised and mention whether the antibody is monoclonal or polyclonal. Please indicate the concentrations used for different experimental procedures.* | | |
| Rabbit anti- MLXIPL/ChREBP | ABclonal Technology | Cat# A7630 |
| Rabbit anti-β-actin | ABclonal Technology | Cat# AC026 |
| Rabbit anti-Progesterone Receptor | ABclonal Technology | Cat# A0321 |
| Rabbit anti-ApoB | ABclonal Technology | Cat# A1330 |
| Rabbit anti-MTTP | ABclonal Technology | Cat# A1746 |
| Rabbit anti-ABCA1 | ABclonal Technology | Cat# A16337 |
| Mouse anti-PKLR | Santa Cruz Biotechnology | Cat# sc-166228 |
| Mouse anti-glucagon | Santa Cruz Biotechnology | Cat# sc-514592 |
| Rabbit anti-SCD1 | Proteintech Group | Cat# 23393-1-AP |
| Rabbit anti-PPARγ | Proteintech Group | Cat# 16643-1-AP |
| Rabbit anti-insulin | Proteintech Group | Cat# 15848-1-AP |
| Rabbit anti-Lamin A/C | Proteintech Group | Cat# 10298-1-AP |
| Rabbit anti-IRS1 | Proteintech Group | Cat# 17509-1-AP |
| Rabbit anti-INSR | Proteintech Group | Cat# 20433-1-AP |
| Rabbit anti-GSK3β | Proteintech Group | Cat# 22104-1-AP |
| Rabbit anti-ABCG1 | Proteintech Group | Cat# 13578-1-AP |
| Goat anti-rabbit IgG | Proteintech Group | Cat# SA00001-2 |
| Goat anti-mouse IgG | Proteintech Group | Cat# SA00001-1 |
| Goat anti-rabbit IgG-Rhodamine | Proteintech Group | Cat# SA00007-2 |
| Rabbit anti-ChREBP | Novus | Cat# NB400-135 |
| Goat anti-mouse IgG-FITC | Sigma-Aldrich | Cat# F0257 |
| Mouse anti-AKT | Cell Signaling Technology | Cat# 5239S |
| Rabbit anti-pi-AKT (Ser473) | Cell Signaling Technology | Cat# 4060L |

| Reagent/Resource | Reference or Source | Identifier or Catalog Number |
|---|---|---|
| Rabbit anti-pi-GSK3β (Ser9) | Cell Signaling Technology | Cat# 5558S |
| Rabbit anti-pi-INSR (Tyr1345) | Cell Signaling Technology | Cat# 3026S |
| Rabbit anti-pi-IRS1 (Ser1101) | Cell Signaling Technology | Cat# 2385 |
| **Oligonucleotides and other sequence-based reagents** *For long lists of oligos or other sequences please refer to the relevant Table(s) or EV Table(s)* | | |
| Sequences of primers for qRT-PCR and promotor analysis | This study | Appendix Table S1 |
| **Chemicals, Enzymes and other reagents** *(e.g., drugs, peptides, recombinant proteins, dyes etc.)* | | |
| 17-β-estradiol (estradiol) | Solarbio | E8140 |
| progesterone | Sigma-Aldrich | P0130 |
| Rosiglitazone | Alexis Biochemicals | ALX-350-125-M025 |
| **Software** *Include version where applicable* | | |
| Adobe Photoshop CS6 | http://www.photoshop.com | |
| GraphPad Prism 8 | http://www.graphpad.com/ | |
| Image J | https://imagej.net/ImageJ | |
| **Other** *(Kits, instrumentation, laboratory equipment, lab ware etc. that are critical for the experimental procedure and do not fit in any of the above categories)* | | |
| Progesterone ELISA kit | Elabscience Biotechnology | E-EL-0154c |
| Insulin ELISA kit | Elabscience Biotechnology | E-EL-M1382 |
| FFA content assay kit | Solarbio | BC0590 |
| TG content assay kit | Wako Chemicals | 290-63701 |
| Cholesterol content assay kit | Applygen Technologies | E1015 |

## Cell culture

HepG2 cells were purchased from ATCC (ATCC, Manassas, VA). Primary hepatocytes were isolated and prepared as described (Zhang et al, 2020a).

Mouse primary hepatocytes were obtained from female mice aged approximately 4–6 weeks using a collagenase perfusion method as previously described (Sun et al, 2017). In brief, mice were anesthetized with chloral hydrate administered by intraperitoneal injection. The inferior vena cava of the mouse was

cannulated with an angiocatheter, the liver was then perfused with a solution consisting of 1 mL heparin (320 U/mL), 40 mL solution I (Krebs solution with 0.1 mmol/L EGTA), and 30 mL solution II (Krebs solution with 2.74 mmol/L $CaCl_2$ and 0.05% collagenase I). The perfused liver was filtered through a 37-μm screen using cold DMEM medium. The collected hepatocytes were then centrifuged three times at 4 °C for 3 min at 50 g and resuspended, and then cultured overnight and subsequently subjected to the designated treatment.

Cells were cultured in DMEM medium. Before treatment, the medium was switched to serum-free and phenol-red-free medium to avoid potential interference from serum hormones or phenol-red.

## Generation of gene knockout mice

Hepatocyte-specific *Pparg*-deficient (h*Pparg* KO) mice and the corresponding control mice (*Pparg*^flox/flox) were generated as described (Zhang et al, 2020b).

*Chrebp*^flox/flox mice (RRID: IMSR_GPT: T009211) were constructed by GemPharmatech (Nanjing, China). Briefly, The Mlxipl gene has 10 transcripts. According to the Mlxipl gene structure, exon 2 of the Mlxipl-201 (ENSMUST00000005507.9) transcript was recommended as the knockout region. The region contains a 107 bp coding sequence, and knocking out the region will result in the disruption of protein function. The brief process is as follows: sgRNA was transcribed in vitro, and the donor vector was constructed. Cas9, sgRNA, and Donor were microinjected into the fertilized eggs of C57BL/6J mice. Fertilized eggs were transplanted to obtain positive F0 mice, which were confirmed using PCR and sequencing. A stable Fl generation mouse model was acquired by mating positive F0 generation mice with C57BL/6J mice. The floxed mice had genes knocked out following mating with mice expressing Cre recombinase, resulting in the loss of function for the target gene in hepatocytes.

To generate hepatocyte-specific *Chrebp*-deficient mice, *Chrebp*^flox/flox mice were crossbred with Alb-Cre transgenic mice. Thus, *Chrebp*^flox/flox/Cre^−/− were named *Chrebp*^flox/flox and *Chrebp*^flox/flox/Cre^+/− were named h*Chrebp* KO mice, respectively.

## In vivo study

Female C57BL/6J wild-type (WT) mice were divided into four groups: CC (mice fed normal chow), CF [mice fed HFrD (Moldiets, MD08040107, 10 kcal% fat, and 70% fructose) from Day 8], PC (pregnant mice fed normal chow) and PF (pregnant mice fed HFrD from Day 8). A portion of the mice (5 mice/group) in the four groups were sacrificed on Day 17. Offspring from the remaining PC and PF groups were maintained and sacrificed at 8 weeks old ($n = 5$).

Female *Chrebp*^flox/flox and h*Chrebp* KO mice were divided into four groups: WPC (*Chrebp*^flox/flox mice fed NC), WPF (*Chrebp*^flox/flox mice fed HFrD after pregnancy), KPC (h*Chrebp* KO mice fed NC), and KPF (h*Chrebp* KO mice fed HFrD after pregnancy). All groups were mated with adult male mice (*Chrebp*^flox/flox) overnight on Day 0 and received corresponding food after pregnancy. A portion of the mice (5 mice in each group) in the four groups were sacrificed on Day 17. Offspring from the remaining pregnant mice were maintained and sacrificed at 8 weeks old ($n = 5$). The offspring

were divided into six groups: fWPC (*Chrebp*flox/flox offspring from WPC group), fWPF (*Chrebp*flox/flox offspring from WPF group), fKPC-WT (*Chrebp*flox/flox offspring from KPC group), fKPC-KO (h*Chrebp* KO offspring from KPC group), fKPF-WT (*Chrebp*flox/flox offspring from KPF group), and fKPF-KO (h*Chrebp* KO offspring from KPF group).

Female mice were mated to male mice on the preceding night and copulatory plugs were checked the following morning, and designed as E1.

GTT and ITT were carried out a few days before the end of study. For GTT, mice were orally administrated with glucose (0.5 g/kg body weight) after 12-h fasting, followed by determination of blood glucose levels at the indicated time points. For ITT, mice were i.p. injected with insulin (1 U/kg body weight) after 6-h fasting, followed by determination of blood glucose levels at the indicated time points.

## Hematoxylin & eosin (HE) and Oil Red O (ORO) staining

Liver and pancreas tissues paraffin sections to conduct HE staining as described (Zhang et al, 2020b). For ORO staining, frozen liver sections were prepared and stained as described (Wang et al, 2022). The sections were photographed by a Leica DM3000 microscope (Wetzlar, Germany).

## Determination of serum insulin and progesterone levels

Mice blood samples were kept for 2 h at room temperature followed by centrifugation for 20 min at $2000 \times g$. Serum levels of insulin and progesterone were measured by the corresponding ELISA kits.

## Western blotting and qRT-PCR

For western blotting, cellular or tissue proteins were extracted and used to determine protein expression. For qRT-PCR, cells or tissues were lysed in TRIzol to extract RNA, and cDNA was synthesized from RNA by reverse transcription followed by real-time PCR to determine the mRNA levels with the primers in Appendix Table S1.

## RNA-seq analysis

RNA-seq and analysis were performed by GENEWIZ Biotechnology Co., Ltd. Briefly, 1 μg total RNA was used for following library preparation. The poly(A) mRNA isolation was performed using Oligo(dT) beads. The mRNA fragmentation was performed using divalent cations and high temperature. Priming was performed using Random Primers. First-strand cDNA and the second-strand cDNA were synthesized. The purified double-stranded cDNA was then treated to repair both ends and add a dA-tailing in one reaction, followed by a T-A ligation to add adapters to both ends. Size selection of Adapter-ligated DNA was then performed using DNA Clean Beads. Each sample was then amplified by PCR using P5 and P7 primers and the PCR products were validated. Then libraries with different indexs were multiplexed and loaded on an Illumina HiSeq/Illumina Novaseq/MGI2000 instrument for sequencing using a $2 \times 150$ paired-end (PE) configuration according to manufacturer's instructions. Bubble chart and volcano plots were plotted by https://www.bioinformatics.com.cn (last

accessed on 10 Oct 2023), an online platform for data analysis and visualization.

## Immunofluorescent staining

Paraffin-embedded liver and pancreas sections from mice were subjected to immunofluorescent staining as described (Ma et al, 2018). Photographs were obtained with immunofluorescence microscopy (Leica DM5000B, Leica Microsystems, Nussloch, Germany).

## Determination of promoter activity

The corresponding promoter was generated as described with the primers in Appendix Table S1 (Zhang et al, 2020a). Cells in 48-well plates were transfected with plasmids for the corresponding promoter and Renilla (for internal normalization). After overnight transfection and treatment, cells were used to determine the activity of firefly and Renilla luciferases using the Dual-Luciferase reporter assay system (Promega, Madison, WI).

## Determination of the lipid content in liver and serum

A piece of liver (~100 mg) or serum was used to determine the TG, FFA, or cholesterol by the corresponding ELISA kits.

## Chromatin immunoprecipitation (ChIP) assay

Cells were subjected to chromatin immunoprecipitation as described (Sun et al, 2017). The precipitated chromatin of the ChREBP promoter region was detected by PCR using the following primers: forward primer (ATGTTAATGAGGCCCGGCTG); reverse primer (CTGTGTCCGAGTCCGAGTCT). The density of each band was quantified with Image J software.

## Human blood RNA extraction

We collected whole blood samples from 12 pregnant women with ($n = 6$) or without ($n = 6$) diabetes mellitus during pregnancy from Tianjin Central Hospital of Gynecology Obstetrics (Nankai University Maternity Hospital). mRNA from human blood was isolated by blood total RNA kit (Zomanbio, China). cDNA was synthesized using the HiScript III-RT SuperMix Kit (Vazyme, China). *Chrebpa* and *Chrebpb* mRNA levels were evaluated by qRT-PCR.

## Data collection and bioinformatics analysis

GEO dataset was downloaded from GEO database (http://www.ncbi.nlm.nih.gov/geo). The GSE17409 dataset contains the subset of data used to generate a healthy donor signature comparing female healthy specimens before pregnancy with respect to female healthy specimens at ninth month of pregnancy. Our analysis contained healthy donors at the 9th month of pregnancy ($n = 3$) and nonpregnant subjects ($n = 5$). We utilized the GEO2R (http://www.ncbi.nlm.nih.gov/geo/geo2r/) statistical tool to calculate and assess the genes that were expressed differently. The resultant DEG dataset was collected and used for further analysis and to generate the volcano map.

## Data analysis

All experiments were repeated at least three times, and representative results were presented. The data are presented as the mean ± SEM and analyzed by Student's t-test or ANOVA with a post hoc test using GraphPad Prism software (San Diego, CA). The differences were considered significant at $P < 0.05$.

## Study approval

The protocol for the animal study was approved by the Ethics Committee of Nankai University (Tianjin, China) and strictly conformed to the Guide for the Care and Use of Laboratory Animals published by the National Institutes of Health (2022-SYDWLL-000573). All patients and healthy controls gave their written informed consent, and the study was approved by Nankai University and conducted in accordance with the Declaration of Helsinki (NKUIRB2022142).

# Data availability

The RNA-seq data have been deposited to Gene Expression Omnibus (GEO) database with the accession code GSE247892 and GSE248051. All data generated in this study are included in this article. Source data are provided with this paper or upon request from the corresponding authors.

# Peer review information

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

## Acknowledgements

This work was supported by the National Natural Science Foundation of China (NSFC) Grants U22A20272, 82173807 to Yajun Duan, 82304503 to Shuang Zhang, Anhui Provincial Natural Science Foundation Grant 2308085QH305 to Shuang Zhang, and the Fundamental Research Funds for the Central Universities (Hefei University of Technology) grant JZ2023HGTB0288 to Shuang Zhang. The ChREBP expression vectors were kindly provided by Professor Xuemei Tong from Shanghai Jiao Tong University. The *Pparg*^flox/flox mice and h*Pparg* KO mice were kindly provided by Professor Shengzhong Duan from Shanghai Jiao Tong University.

## Author contributions

**Jiaqi Li**: Conceptualization; Validation; Investigation; Methodology; Writing—original draft. **Shuang Zhang**: Validation; Investigation; Methodology; Writing—review and editing. **Yuyao Sun**: Validation; Investigation. **Jian Li**: Validation; Investigation. **Zian Feng**: Software; Formal analysis. **Huaxin Li**: Validation; Investigation. **Mengxue Zhang**: Validation; Investigation. **Tengteng Yan**: Validation; Investigation. **Jihong Han**: Conceptualization; Supervision; Investigation; Methodology; Project administration. **Yajun Duan**: Conceptualization; Supervision; Funding acquisition; Investigation; Methodology; Project administration; Writing—review and editing.

## Disclosure and competing interests statement

The authors declare no competing interests.

# Expanded View Figures

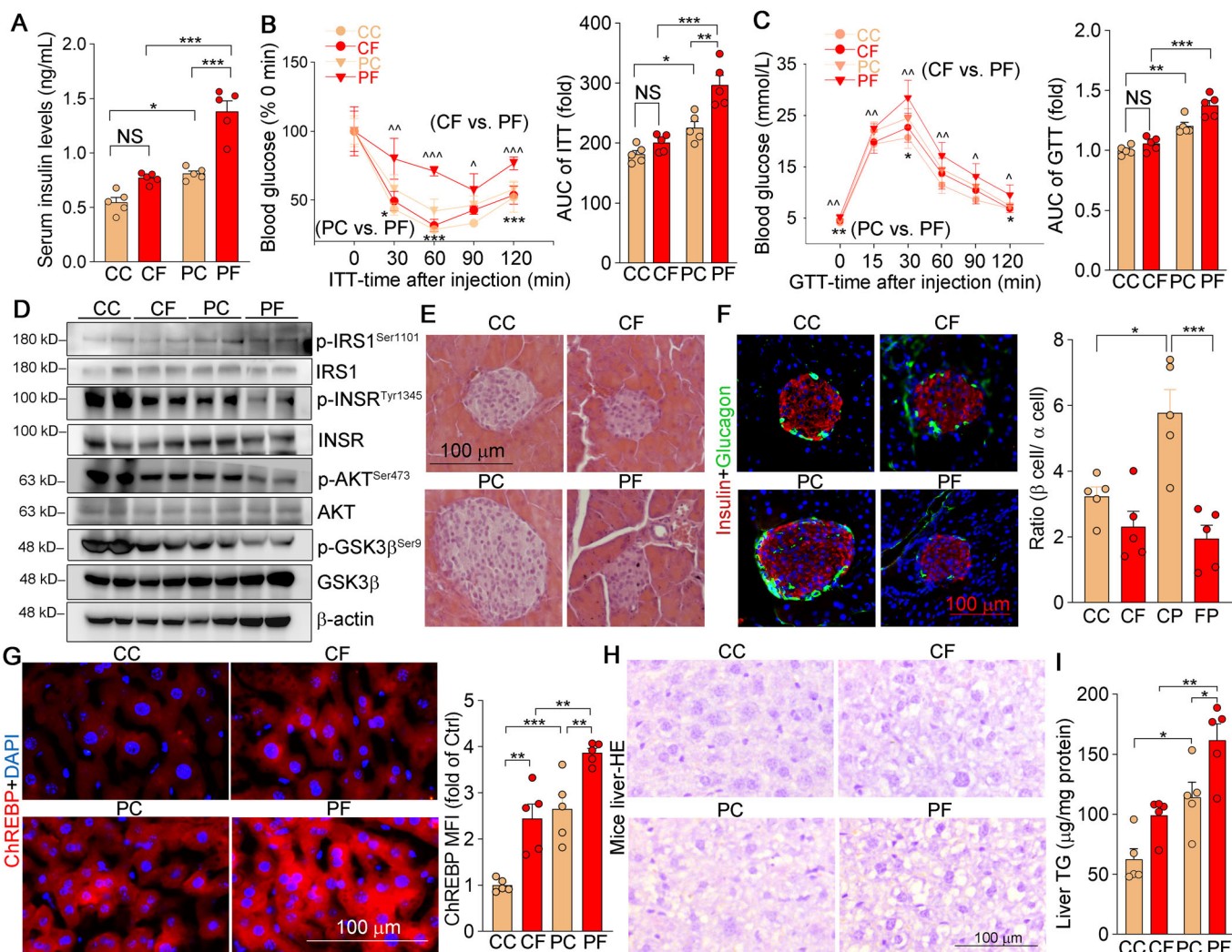

**Figure EV1. HFrD aggravates glucose and lipid metabolism disorders in maternal mice.**

The mice in Fig. 1E were used to complete following assays: (A–C) Serum insulin level (A), ITT (B) or GTT (C) assay at the E14 or E15, and quantitation of areas under curves (AUC). (D) Expression of protein related to insulin signaling pathway in mouse liver was determined by Western blot. (E) HE staining of mice pancreas sections. (F) Immunofluorescent staining with anti-glucagon (green) or anti-insulin (red) antibodies of mice pancreas sections, and the ratio of beta cell area (insulin-positive) to alpha cell area (glucagon-positive) was calculated, $n = 5$. (G) The protein expression of ChREBP was determined by immunofluorescent staining with quantification of the mean immunofluorescence intensity (MFI) of images. (H) HE staining of liver sections. (I) triglyceride (TG) quantification in the liver. $n = 5$. Data information: All graphs are represented as Mean ± SEM, $n$: biological replicates. Two-way ANOVA followed by Tukey's multiple comparisons test was used. $*P < 0.05$, $**P < 0.01$, $***P < 0.001$, NS: no significance vs. indicated group. Source data are available online for this figure.

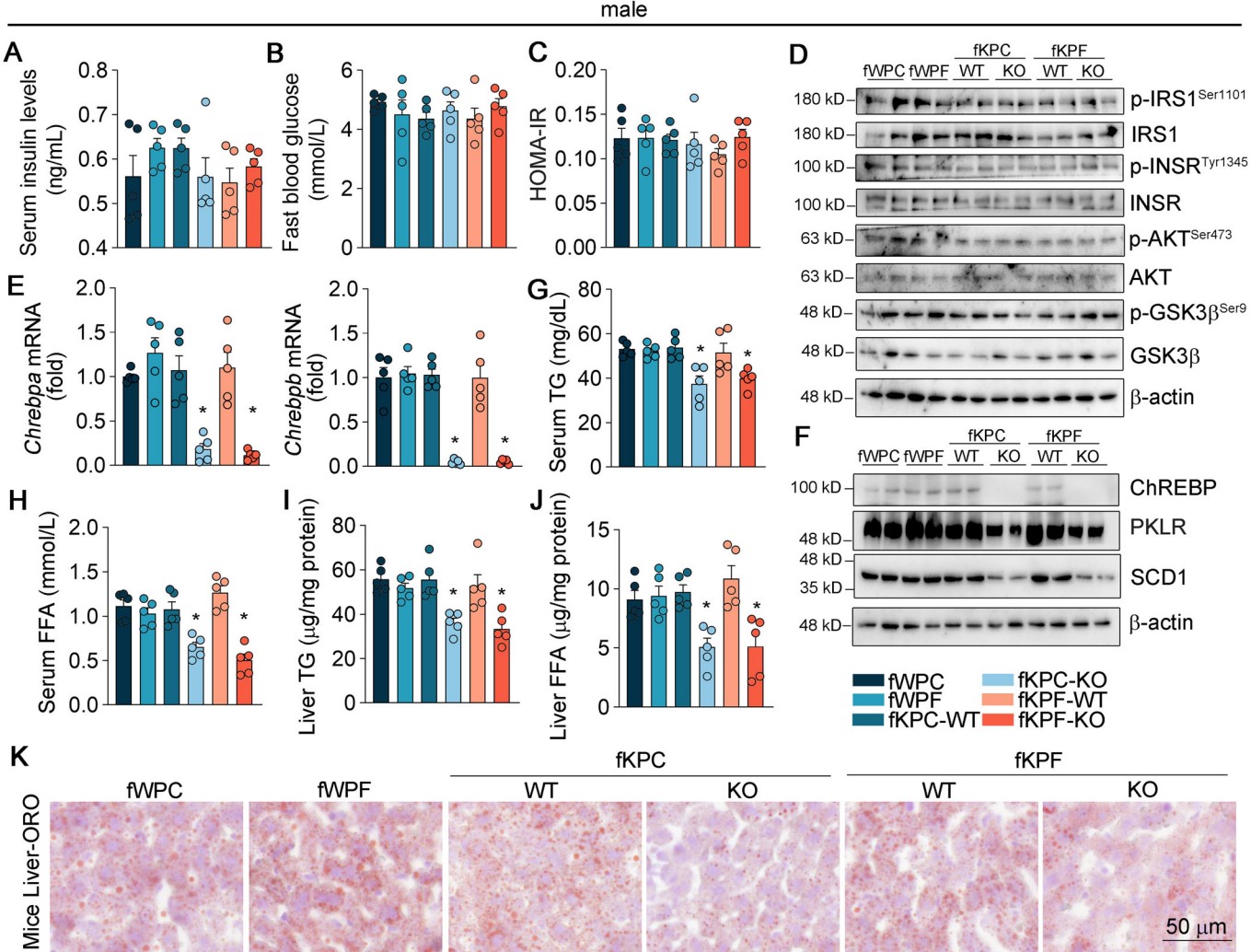

**Figure EV2.  Hepatic ChREBP deficiency did not improve HFrD-impaired the glucose and lipid homeostasis of male offspring.**

The male offspring in Fig. 7A were used to complete following assays: (**A–C**) Serum insulin levels (**A**), fast blood glucose (**B**) and HOMA-IR index (**C**) of male offspring. (**D,F**) Expression of protein related to insulin signaling pathway (**D**) and ChREBP and its downstream target protein (**F**) was determined by Western blot in the liver from male offspring. (**E**) Expression of *Chrebpa* and *Chrebpb* mRNA in the liver from male offspring was determined by qRT-PCR. (**G–J**) The serum (**G** and **H**) and liver (**I** and **J**) were conducted triglyceride (TG) and free fatty acid (FFA) quantitative analysis. (**I**) Oil Red O (ORO) staining of liver sections (**K**). $n = 5$. Data information: All graphs are represented as Mean ± SEM, *n*: biological replicates. One-way ANOVA followed by Tukey's multiple comparisons test was used. *$P < 0.05$ vs. fWPC group. Source data are available online for this figure.

