## [Peer Review File · EMBO Reports]

Liver ChREBP deficiency inhibits fructose-induced insulin resistance in pregnant mice and female offspring

Jiaqi Li, Shuang Zhang, Yuyao Sun, Jian Li, Zian Feng, Huaxin Li, Mengxue Zhang, Tengting Yan, Jihong Han, and Yajun Duan

Corresponding author(s): Yajun Duan (yajunduan@ustc.edu.cn) , Jihong Han (jihonghan2008@nankai.edu.cn)

Review Timeline:

Submission Date:	7th Jul 23
Editorial Decision:	8th Sep 23
Revision Received:	30th Nov 23
Editorial Decision:	31st Jan 24
Revision Received:	18th Feb 24
Accepted:	29th Feb 24

Editor: Deniz Senyilmaz Tiebe

Transaction Report:

Dear Prof. Duan,

Thank you for the submission of your research manuscript to our journal, which was now seen by three referees, whose reports are copied below.

My apologies for this unusual delay in getting back to you. It took longer than anticipated to receive the full set of referee reports.

The referees express interest in the proposed role of liver ChREBP in fructose induced insulin resistance in the mother and the female offspring. However, they also raise significant concerns that need to be addressed to consider publication here.

Should you be able to address all referee concerns, we would like to invite you to submit a revised manuscript. Please revise your manuscript with the understanding that the referee concerns (as in their reports) must be fully addressed and their suggestions taken on board. Please address all referee concerns in a complete point-by-point response. Acceptance of the manuscript will depend on a positive outcome of a second round of review. It is EMBO reports policy to allow a single round of major experimental revision only and acceptance or rejection of the manuscript will therefore depend on the completeness of your responses included in the next, final version of the manuscript.

We realize that it is difficult to revise to a specific deadline. In the interest of protecting the conceptual advance provided by the work, we recommend a revision within 3 months. Please discuss the revision progress ahead of this time with me if you require more time to complete the revisions, or if you have questions or comments regarding the revision (also by video chat).

1. A data availability section providing access to data deposited in public databases is missing (where applicable).
2. Your manuscript contains statistics and error bars based on $n=2$. Please use scatter plots in these cases.

You can submit the revision either as a Scientific Report or as a Research Article. For Scientific Reports, the revised manuscript can contain up to 5 main figures and 5 Expanded View figures, and it should not exceed 27000 characters. If the revision leads to a manuscript with more than 5 main figures it will be published as a Research Article. In this case the Results and Discussion section should be separate. If a Scientific Report is submitted, these sections have to be combined. This will help to shorten the manuscript text by eliminating some redundancy that is inevitable when discussing the same experiments twice. In either case, all materials and methods should be included in the main manuscript file.

4) a .docx formatted letter INCLUDING the reviewers' reports and your detailed point-by-point responses to their comments. As part of the EMBO publication's Transparent Editorial Process, EMBO reports publishes online a Review Process File (RPF) to accompany accepted manuscripts. This File will be published in conjunction with your paper and will include the referee reports, your point-by-point response and all pertinent correspondence relating to the manuscript.

<https://www.embopress.org/page/journal/14693178/authorguide#transparentprocess>

5) a complete author checklist, which you can download from our author guidelines <https://www.embopress.org/page/journal/14693178/authorguide>. Please insert information in the checklist that is also reflected in the manuscript. The completed author checklist will also be part of the RPF.

6) Please note that all corresponding authors are required to supply an ORCID ID for their name upon submission of a revised manuscript (<<https://orcid.org/>>). Please find instructions on how to link your ORCID ID to your account in our manuscript tracking system in our Author guidelines <<https://www.embopress.org/page/journal/14693178/authorguide#authorshipguidelines>>

Additional information on source data and instruction on how to label the files are available: <https://www.embopress.org/page/journal/14693178/authorguide#sourcedata>

9) Our journal encourages inclusion of *data citations in the reference list* to directly cite datasets that were re-used and obtained from public databases. Data citations in the article text are distinct from normal bibliographical citations and should directly link to the database records from which the data can be accessed. In the main text, data citations are formatted as follows: "Data ref: Smith et al, 2001" or "Data ref: NCBI Sequence Read Archive PRJNA342805, 2017". In the Reference list, data citations must be labeled with "[DATASET]". A data reference must provide the database name, accession number/identifiers and a resolvable link to the landing page from which the data can be accessed at the end of the reference. Further instructions are available at <http://www.embopress.org/page/journal/14693178/authorguide#referencesformat>

- the name of the statistical test used to generate error bars and P values,
- the number (n) of independent experiments (please specify technical or biological replicates) underlying each data point,
- the nature of the bars and error bars (s.d., s.e.m.),
- If the data are obtained from n Program fragment delivered error `Can't locate object method "less" via package "than" (perhaps you forgot to load "than"?) at //ejpvfs23/sites23b/embo_www/letters/embo_decision_revise_and_review.txt line 56.' 2, use scatter blots showing the individual data points.

12) Please also note our reference format:

I look forward to seeing a revised version of your manuscript when it is ready. Please let me know if you have questions or comments regarding the revision.

Kind regards,

Deniz Senyilmaz Tiebe

Deniz Senyilmaz Tiebe, PhD
Editor
EMBO Reports

Referee #1:

The manuscript by Jiaqi Li entitled "Liver ChREBP deficiency inhibits fructose-induced insulin resistance in mother and female offspring" describes the key role of the transcription factor ChREBP in insulin resistance during pregnancy. Combining several experimental approaches, including genetically-engineered mouse models, pharmacological treatments, molecular analysis, in vivo AAV experiments, the authors convincingly decipher the molecular link between ChREBP-progesterone-PPAR γ to trigger insulin resistance during pregnancy. Moreover, the authors show in this manuscript that female offspring from IP pregnant mothers are also sensitive to develop glucose homeostasis impairments, which is dependent on hepatic expression of ChREBP.

This is a nice study that increases our knowledge about the function of ChREBP in carbohydrate metabolism. In addition, since the data to date concerning the physiological/pathophysiological role of ChREBP in glucose homeostasis using mouse models is controversial, this study shed light on potential explanations of why modulating ChREBP activity may have beneficial and/or deleterious consequences depending on the physiological status.

The rescue experiment with AAV-MTTP is also convincing and suggest a role for cholesterol trafficking in IR during pregnancy.

An important conclusion of the paper is the sexual dimorphism effect of liver-specific deficiency observed on ChREBP-mediated insulin resistance improvement in mice.

As presented in the manuscript, the data are convincing. The mechanistic link between progesterone, PPAR γ and the increase in ChREBP expression, which in turn increases the level of cholesterol that will then increase progesterone synthesis, is well addressed.

There are, however, several points that should be addressed.

There is an inversion in the main text between figure 2E (oGTT) and 2F (ITT).

The quality of immunofluorescence (IF) images should be improved.

What are the fasting insulin levels in the different models used in the study? Concerning the pancreatic section, it is difficult to conclude based on a single IF image. Beta-cell mass, or at least alpha to beta cell ratio, should be measured in the wt and hChREBP ko mice and offspring.

Referee #2:

The manuscript by Li et al describe a novel feedback loop of progesterone-liver ChREBP and gestational diabetes as well as linked metabolic complication in offspring. The questions that are followed up in the paper are clearly stated and the experiments to investigate the stated hypotheses are well designed and well presented. Both questions posed and findings presented here are novel.

Major comments :

1. Increase in ChREBP, progesterone and liver cholesterol export are physiological adaptations to pregnancy. It is important to show developmental parameters in offspring. Eg. body weight, skull size, tibial length, organ size and H&E staining. This is to

understand whether deletion of ChREBP or MTT could have caused issues with the normal development of the offspring leading to metabolic problems later in life.

2. A characterization of LO2 (normal human hepatocytes) to show the lack of contamination or not. Caution has been raised on LO2 for contamination.

According to <https://aasldpubs.onlinelibrary.wiley.com/doi/10.1002/hep.32730>

Letter to the Editor: LO2, a misidentified cell line: Some data should be interpreted with caution.

Minor

1. Figure 1 A please indicate in the figure legend the origin tissue for these mRNA expression data.

2. Figure 1 D. Please indicate in the figure legend the tissue for this Immunofluorescent staining.

3. The authors measured ChREBP in whole blood from humans on pregnancy. To link the mouse and human data could be useful to measure ChREBP in whole blood from mice. This will also help to check whether whole blood ChREBP could be a marker for increase in liver ChREBP during pregnancy.

Referee #3:

The authors seek to investigate the role of hepatic ChREBP with or without fructose supplementation in pregnancy associated dysmetabolism and insulin resistance using a sophisticated combination of genetic models. The authors provide some interesting and novel data that hepatic ChREBP may impact risk of dysmetabolism in progeny following overfeeding dams with a high-fructose diet. The authors also provide novel evidence of a relationship between hepatic ChREBP and progesterone and to a lesser extent estradiol. However, the significance of this is not experimentally addressed. Moreover, evidence in this manuscript indicates that this is not an important mechanism regulating pregnancy induced insulin resistance, contrary to the authors' statements. There are a large number of substantial methodological issues that cloud interpretation of the experiments and conclusions in this manuscript, some of which are detailed below. There are multiple instances where stated conclusions are not supported by the data. As a result, despite some novelty, this manuscript will require extensive additional experimentation and revision and reconsideration of major conclusions before it can be adequately evaluated for publication.

Major Issues:

1. Supplemental Fig2A: There is an increase in the protein levels of some ChREBP targets (Pklr and SCD1) but not others (Fasn, Acly, Elovl6) over the course of pregnancy. The expression of all of these factors can be regulated by additional transcription factors, but the lack of concordance does not strongly support the conclusion that hepatic ChREBP activity increases over the course of pregnancy and undermines a key construct of this manuscript.

2. Fig 1D and elsewhere and related to Point 1 above: The authors should measure the expression of distinct ChREBP isoforms by qPCR. Expression of ChREBP-beta tends to be a good index of tissue ChREBP activity whereas expression of ChREBP-alpha both at the gene and protein level is not a strong marker for the ChREBP activity state because its transcriptional activity requires allosteric activation by carbohydrate metabolites. An increase in ChREBP-alpha mRNA and protein is not necessarily indicative of increased ChREBP activity. Also, according to the literature, fructose feeding markedly increases the mRNA expression of the ChREBP-beta isoform which is not easily detected by Western Blot. Fructose feeding has variable effect on the ChREBP-alpha isoform, an isoform which is readily detectable on Western Blot, but is not indicative of ChREBP activity. Whether ChREBP is activated in the liver or elsewhere by pregnancy is not clear based on the data provided in this manuscript.

3. Figure 2: Liver ChREBP KO did not effect serum insulin, GTT, or ITT in KPC vs WPC indicating that liver ChREBP has little or no role in normal insulin resistance of pregnancy. Therefore, the motivation and interpretation of experiments studying the role of ChREBP in regulating progesterone or other factors that contribute to insulin resistance of pregnancy is not clear. Moreover, if taken at face value, this would suggest that progesterone, which does decrease in ChREBP KO mice (Fig 3C) does not play a significant role in pregnancy-induced insulin resistance.

4. The authors invoke a complicated mechanism regarding the effects of fructose and ChREBP on progesterone but provide little experimental data to directly investigating this hypothesis. This remains largely speculative.

Other Issues:

5. Fig 1A and B: Few methods are provided regarding these analyses. Panel A and B suggests these are blood samples. Are ChREBP levels in blood relevant to the hypotheses and model proposed in the rest of the paper? In panel B, where are these samples from? Is this a pre-registered study? Is this IRB approved? There is no discussion of this in the methods.

6. Fig 1E: The authors use an anti-ChREBP AB (ABclonal A7630) which has not previously been published and is not commonly used in the field. The representative blot on the vendors website shows a very prominent non-specific band and is not typical of a high-quality blot for ChREBP. While the authors show some evidence that this antibody can detect ChREBP by immunoblot (Panel 7G), the non-specific binding might preclude its use for IHC. Use for IHC should be validated using KO tissues.

7. The authors provide an RRID number regarding the ChREBP floxed mice but search of the RRID database turns up no information for this mouse. More information regarding this floxed mouse, how it was generated, which exons were floxed, how it was validated are required to evaluate this manuscript.
8. Supp Fig 3C. Although there appears to be a strong effect of fructose to affect INSR phosphorylation in pregnancy, there was no clear effect on downstream signaling despite statements suggesting otherwise in the text.
9. Figure 11: Again, there is no consistent effect of pregnancy (PC vs CC) on the protein levels of ChREBP targets. Also, fructose did not increase the protein levels of ChREBP targets in pregnant mice (PF compared to CF) which suggests that the effects of fructose and pregnancy are not synergistic on hepatic ChREBP activity.
10. Figure 2: Liver ChREBP KO did reduce fructose-induced insulin resistance. However, this is not novel and has previously been published by other groups, though not in the setting of pregnancy. The authors point to an anomalous paper by Jois et al (2017) that suggested that hepatic ChREBP KO impairs glucose homeostasis, but multiple other independent groups have published to the contrary and the results reported by Jois may depend on the nature of the gene deletion performed by these investigators.

Response to the comments from the Reviewer #1

Referee #1:

The manuscript by Jiaqi Li entitled "Liver ChREBP deficiency inhibits fructose-induced insulin resistance in mother and female offspring" describes the key role of the transcription factor ChREBP in insulin resistance during pregnancy. Combining several experimental approaches, including genetically-engineered mouse models, pharmacological treatments, molecular analysis, in vivo AAV experiments, the authors convincingly decipher the molecular link between ChREBP-progesterone-PPAR γ to trigger insulin resistance during pregnancy. Moreover, the authors show in this manuscript that female offspring from IP pregnant mothers are also sensitive to develop glucose homeostasis impairments, which is dependent on hepatic expression of ChREBP.

This is a nice study that increases our knowledge about the function of ChREBP in carbohydrate metabolism. In addition, since the data to date concerning the physiological/pathophysiological role of ChREBP in glucose homeostasis using mouse models is controversial, this study shed light on potential explanations of why modulating ChREBP activity may have beneficial and/or deleterious consequences depending on the physiological status.

The rescue experiment with AAV-MTTP is also convincing and suggest a role for cholesterol trafficking in IR during pregnancy.

An important conclusion of the paper is the sexual dimorphism effect of liver-specific deficiency observed on ChREBP-mediated insulin resistance improvement in mice.

As presented in the manuscript, the data are convincing. The mechanistic link between progesterone, PPAR γ and the increase in ChREBP expression, which in turn increases the level of cholesterol that will then increase progesterone synthesis, is well addressed.

There are, however, several points that should be addressed:

1. There is an inversion in the main text between figure 2E (oGTT) and 2F (ITT).
2. The quality of immunofluorescence (IF) images should be improved.
3. What are the fasting insulin levels in the different models used in the study? Concerning the pancreatic section, it is difficult to conclude based on a single IF image. Beta-cell mass, or at least alpha to beta cell ratio, should be measured in the wt and hChREBP ko mice and offspring.

Dear reviewer,

Thank you for the review of our manuscript and for your constructive recommendations and comments, which have greatly assisted us in improving the manuscript. We have conducted additional experiments and heavily revised our manuscript. The revised segments of the manuscript are indicated in **YELLOW**. The point-by-point responses to comments are listed below. We anticipate that your comments have been addressed accurately and completely.

Comment 1: There is an inversion in the main text between figure 2E (oGTT) and 2F (ITT).

Response 1: Thank you for this helpful comment. We have reversed the order of oGTT and ITT in the revised **NEW Figure 2** to make sure that the text aligned with the figures: the results of oGTT are shown in **NEW Figure 2F**, and the results of ITT are shown in **NEW**

Figure 2E.

.....
Comment 2: The quality of immunofluorescence (IF) images should be improved.

Response 2: Thank you for this recommendation. To enhance the quality of IF images, we have repeated all immunofluorescence experiments (**NEW Figure 1C, Figure 2H and 2I, Supplemental Figure 3F and 3G, Supplemental Figure 10**) by augmenting the experimental conditions, and meticulously adhering to established protocols to acquire results with higher quality.

(1) NEW Figure 1:

Figure for referee with unpublished data and its description has been removed upon request by the authors.

(2) NEW Figure 2:

Figure for referee with unpublished data and its description has been removed upon request by the authors.

(3) NEW Supplemental Figure 3:

Figure for referee with unpublished data and its description has been removed upon request by the authors.

(4) NEW Supplemental Figure 10:

Figure for referee with unpublished data and its description has been removed upon request by the authors.

.....
Comment 3: What are the fasting insulin levels in the different models used in the study? Concerning the pancreatic section, it is difficult to conclude based on a single IF image. Beta-cell mass, or at least alpha to beta cell ratio, should be measured in the wt and h*Chrebp* ko mice and offspring.

Response 3: Thank you for this recommendation.

(1) Fasting insulin levels:

We have assessed the fasting insulin levels throughout all the models employed in our study and included these findings in the manuscript. The results are also summarized in the **Figure** below:

Figure for referee with unpublished data and its description has been removed upon request by the authors.

Data Shown to Reviewer:

(A) The E0 group was unmated and defined on Day 0 of embryonic development. The remaining three groups were mated with adult male mice overnight, and pregnant mice were sacrificed on embryonic Day 2, Day 11, and Day 17, known as E2, E11, and E17 groups. Characterization of serum insulin. n = 5, ***P<0.001 compared to the E0 group. (NEW Supplemental Figure 1A).

(B) Female C57BL/6J wild-type mice were categorized into four groups: CC group, C57BL/6J mice receiving NC; CF group, C57BL/6J mice receiving HFrD from Day 8 to Day 17 of pregnancy; PC group, pregnant mice receiving NC; PF group, pregnant mice receiving HFrD from Day 8 to Day 17 of pregnancy. Calculation of serum insulin. n = 5, NS, no significance. *P<0.05, ***P<0.001 compared to the indicated group. (NEW Supplemental Figure 3A).

(C) Female *Chrebp*^{flox/flox} and hepatocyte-specific *Chrebp* KO (h*Chrebp* KO) mice were mated with adult male mice overnight on Day 0 and separated into four groups receiving NC or HFrD from Day 0 to Day 17: WPC (*Chrebp*^{flox/flox} mice fed NC), WPF (*Chrebp*^{flox/flox} mice fed HFrD), KPC (h*Chrebp* KO mice fed NC), and KPF (h*Chrebp* KO mice fed HFrD). All mice were sacrificed on Day 17. Quantification of serum insulin. n = 10, *P<0.05, ***P<0.001 compared to the indicated group. (NEW Figure 2B).

(D) Female *Chrebp*^{flox/flox} and h*Chrebp* KO mice were given AAV-Ctrl or AAV-Mttp via the tail vein. Subsequently, they were mated with adult male mice overnight and subjected to HFrD from Day 0 to Day 17. Quantification of serum insulin. n = 5, ***P<0.001 compared to the indicated group. (NEW Figure 4K).

(E) Offspring of the pregnant mice in the PC or PF groups (similar to the experiment in Figure 1E) were respectively defined as the fPC or fPF group and maintained until the age of 8 weeks (sexual maturity). Quantification of serum insulin. n = 5 to 6, ***P<0.001 compared to the indicated group. (NEW Figure 6B)

(F) Offspring from the pregnant mice in the WPC, WPF, KPC, and KPF groups (similar to the experiment in Figure 2A) were respectively defined as the fWPC, fWPF, fKPC, and fKPF groups and maintained until the age of 8 weeks (sexual maturity). n = 5, ***P<0.001 compared to the fWPC group. (NEW Figure 7B, Supplemental Figure 11A).

(2) Pancreatic section analysis:

We determined the beta-to-alpha cell ratio in both wild-type and hepatic ChREBP-deficient mice, and their offspring. We have included these data in the manuscript (NEW Figure 2H, Figure S3E, Figure S10). The findings are summarized in the Figure below:

Figure for referee with unpublished data and its description has been removed upon request by the authors.

Figure for referee with unpublished data and its description has been removed upon request by the authors.

Figure for referee with unpublished data and its description has been removed upon request by the authors.

Response to the comments from the Reviewer #2

Referee #2:

The manuscript by Li et al describe a novel feedback loop of progesterone-liver ChREBP and gestational diabetes as well as linked metabolic complication in offspring. The questions that are followed up in the paper are clearly stated and the experiments to investigate the stated hypotheses are well designed and well presented. Both questions posed and findings presented here are novel.

Major comments:

1. Increase in ChREBP, progesterone and liver cholesterol export are physiological adaptations to pregnancy. It is important to show developmental parameters in offspring. Eg. body weight, skull size, tibial length, organ size and H&E staining. This is to understand whether deletion of ChREBP or MTT could have caused issues with the normal development of the offspring leading to metabolic problems later in life.
2. A characterization of LO2 (normal human hepatocytes) to show the lack of contamination or not. Caution has been raised on LO2 for contamination.

According to <https://aasldpubs.onlinelibrary.wiley.com/doi/10.1002/hep.32730>

Letter to the Editor: LO2, a misidentified cell line: Some data should be interpreted with caution.

Minor comments:

1. Figure 1 A please indicate in the figure legend the origin tissue for these mRNA expression data.
2. Figure 1 D. Please indicate in the figure legend the tissue for this Immunofluorescent staining.
3. The authors measured ChREBP in whole blood from humans on pregnancy. To link the mouse and human data could be useful to measure ChREBP in whole blood from mice. This will also help to check whether whole blood ChREBP could be a marker for increase in liver ChREBP during pregnancy.

Dear reviewer,

Thank you for the review of our manuscript and for your constructive recommendations and comments, which have greatly assisted us in improving the manuscript. We have conducted additional experiments and heavily revised our manuscript. The revised segments of the manuscript are indicated in **YELLOW**. The point-by-point responses to comments are listed below. We anticipate that your comments have been addressed accurately and completely.

Major comments :

Comment 1: Increase in ChREBP, progesterone and liver cholesterol export are physiological adaptations to pregnancy. It is important to show developmental parameters in offspring. Eg. body weight, skull size, tibial length, organ size and H&E staining. This is to understand whether deletion of ChREBP or MTT could have caused issues with the normal development of the offspring leading to metabolic problems later in life.

Response 1: Thank you for this insightful comment. To verify our answers to your questions, we have collected the following results.

(1) Firstly, we examined the data from 8-week-old offspring mice in the NEW Figure 7A:

Pale liver was observed in the female offspring mice from mothers administered a high-fructose diet (**B**), potentially due to the activation of the "progesterone-ChREBP" pathway in female offspring linked to excess maternal fructose intake. There were no significant alterations in the body weight, liver weight, and growth hormone level in the offspring mice (**C, D**).

Figure for referee with unpublished data and its description has been removed upon request by the authors.

Data Shown to Reviewer: Liver morphology, body weight, and liver weight of 8-week-old female and male offspring.

(A) Offspring from the pregnant mice in the WPC, WPF, KPC, and KPF groups (similar to the experiment in Figure 2A) were, defined as the fWPC, fWPF, fKPC, and fKPF groups, respectively, and maintained until the age of 8 weeks (sexual maturity).

(B) Liver morphology of 8-week-old female and male offspring.

(C) Body weight and liver weight of 8-week-old female and male offspring. n = 5.

(D) Serum growth hormone levels of 8-week-old female and male offspring. n = 5.

(2) We next assessed the bone dimensions of embryonic mice (E17):

The results suggested that hepatic ChREBP knockout does not impact the offspring's skull width, skull length, and tibia length at embryonic day 17, regardless of whether the mothers

were administered a regular diet or a high-fructose diet.

Figure for referee with unpublished data and its description has been removed upon request by the authors.

Data Shown to Reviewer: Skull length, skull width, and tibia length of embryonic day 17 (E17) offspring.

(A) Female *Chrebp*^{flox/flox} and hepatocyte-specific *Chrebp* KO (h*Chrebp* KO) mice were mated with adult male mice overnight on Day 0 and separated into four groups receiving NC or HFrD from Day 0 to Day 17, and fetal mice were obtained at the embryonic day 17 (E17): fWPC (*Chrebp*^{flox/flox} mice fed NC), fWPF (*Chrebp*^{flox/flox} mice fed HFrD), fKPC (h*Chrebp* KO mice fed NC), and fKPF (h*Chrebp* KO mice fed HFrD).

(B) Skull length, skull width, and tibia length measurements of embryonic day 17 (E17) offspring. n = 5.

(3) The effects of overexpression of hepatic MTTP:

Hepatic overexpression of *Mttp* did not impact the offspring's skull width, skull length, and tibia length at embryonic day 17, irrespective of mothers being fed a regular diet or a high-fructose diet.

Figure for referee with unpublished data and its description has been removed upon request by the authors.

Data Shown to Reviewer: Skull length, skull width, and tibia length of embryonic day 17 (E17) offspring.

(A) Female *Chrebp*^{flox/flox} (wild type) mice were given AAV-Ctrl or AAV-Mttp via the tail vein. Subsequently, they were mated with adult male mice overnight and given NC or HFrD from Day 0 to Day 17.

(B) Skull length, skull width, and tibia length measurements at embryonic day 17 (E17) offspring.
n = 5.

(4) Hepatic ChREBP deficiency did not impact the basic characteristics of both female and male mice:

To give further evidence, we obtained data from hepatic ChREBP knockout mice (those not exposed to fructose). These mice exhibited no significant alterations in development compared to their wild-type counterparts. This supports the idea that these genetic modifications are unlikely to influence growth and development significantly.

Figure for referee with unpublished data and its description has been removed upon request by the authors.

Data Shown to Reviewer: Skull length, skull width, tibia length, tissue structure, and body morphology of 8-week-old offspring.

(A, B) Skull length, skull width, and tibia length measurements of 8-week-old female (A) and male (B) offspring.

(C, D) Microscopic examination of tissue structure via HE staining of 8-week-old female (C) and male (D) offspring.

(E, F) Body morphology of 8-week-old female (E) and male (F) offspring, n = 5.

Consistently, Lantz et al.'s study suggests that a maternal high-fructose diet causes offspring to have a non-significant change in growth efficiency and BMI efficiency prior to 18 weeks of age, with non-significant changes in body weight before 10 weeks of age compared to the control group (Arentson-Lantz *et al*, 2016). Therefore, according to the above results, the metabolic alterations in the offspring are unlikely to be attributed to metabolic differences.

.....
Comment 2: A characterization of LO2 (normal human hepatocytes) to show the lack of contamination or not. Caution has been raised on LO2 for contamination.

According to <https://aasldpubs.onlinelibrary.wiley.com/doi/10.1002/hep.32730>

Letter to the Editor: LO2, a misidentified cell line: Some data should be interpreted with caution.

Response 2: Thank you for this well-considered comment. We have replaced the LO2 cell line with either primary mouse hepatocytes or HepG2 cells. The findings are summarized in the **Figure** below:

(1) *NEW Figure 5:*

Figure for referee with unpublished data and its description has been removed upon request by the authors.

(2) *NEW Supplemental Figure 5:*

Figure for referee with unpublished data and its description has been removed upon request by the authors.

(3) NEW Supplemental Figure 8:

Figure for referee with unpublished data and its description has been removed upon request by the authors.

(4) NEW Supplemental Figure 12:

Figure for referee with unpublished data and its description has been removed upon request by the authors.

.....
Minor comments :

Comment 3: Figure 1 A please indicate in the figure legend the origin tissue for these mRNA expression data.

Response 3: Thank you for this insightful comment. We have updated the legend for **Figure 1A** to specify that the mRNA expression data presented in this figure are derived from peripheral blood mononuclear cells, and moved it from **Figure 1A** to **NEW Supplemental Figure 13A**. (*Line 157-158 of Page 17 in supplemental data*)

.....
Comment 4: Figure 1 D. Please indicate in the figure legend the tissue for this Immunofluorescent staining.

Response 4: Thank you for this helpful suggestion. We have updated the figure legend for **Figure 1D** (moved to **NEW Figure 1C** in the revised manuscript) to stipulate that the immunofluorescent staining data originate from the mouse liver. (*Line 758 of Page 34 in manuscript*)

.....
Comment 5: The authors measured ChREBP in whole blood from humans on pregnancy. To link the mouse and human data could be useful to measure ChREBP in whole blood from mice. This will also help to check whether whole blood ChREBP could be a marker for increase in liver ChREBP during pregnancy.

Response 5: Thank you for this helpful comment. We have conducted experiments to assess ChREBP levels in whole blood derived from mice (**NEW Supplemental Figure 13B and 13C**). We are pleased to report that the results obtained from the mouse blood samples are similar to the findings from our human data. Additionally, this finding supports the hypothesis that whole-blood ChREBP may be a marker for increased liver ChREBP during pregnancy. In response to this important recommendation, we have included a section within the discussion

highlighting the significance of measuring ChREBP mRNA levels in blood. We highlight that this approach provides insight into the potential link between mouse and human data and serves as a valuable avenue for investigating if whole-blood ChREBP can be a marker for elevated liver ChREBP during pregnancy. (*Line 406-416 of Page 19 and Line 417-427 of Page 20 in manuscript*)

Figure for referee with unpublished data and its description has been removed upon request by the authors.

Response to the comments from the Reviewer #3

Referee #3:

The authors seek to investigate the role of hepatic ChREBP with or without fructose supplementation in pregnancy associated dysmetabolism and insulin resistance using a sophisticated combination of genetic models. The authors provide some interesting and novel data that hepatic ChREBP may impact risk of dysmetabolism in progeny following overfeeding dams with a high-fructose diet. The authors also provide novel evidence of a relationship between hepatic ChREBP and progesterone and to a lesser extent estradiol. However, the significance of this is not experimentally addressed. Moreover, evidence in this manuscript indicates that this is not an important mechanism regulating pregnancy induced insulin resistance, contrary to the authors' statements. There are a large number of substantial methodological issues that cloud interpretation of the experiments and conclusions in this manuscript, some of which are detailed below. There are multiple instances where stated conclusions are not supported by the data. As a result, despite some novelty, this manuscript will require extensive additional experimentation and revision and reconsideration of major conclusions before it can be adequately evaluated for publication.

Major Issues:

1. Supplemental Fig2A: There is an increase in the protein levels of some ChREBP targets (Pklr and SCD1) but not others (Fasn, Acly, Elovl6) over the course of pregnancy. The expression of all of these factors can be regulated by additional transcription factors, but the lack of concordance does not strongly support the conclusion that hepatic ChREBP activity increases over the course of pregnancy and undermines a key construct of this manuscript.
2. Fig 1D and elsewhere and related to Point 1 above: The authors should measure the expression of distinct ChREBP isoforms by qPCR. Expression of ChREBP-beta tends to be a good index of tissue ChREBP activity whereas expression of ChREBP-alpha both at the gene and protein level is not a strong marker for the ChREBP activity state because its transcriptional activity requires allosteric activation by carbohydrate metabolites. An increase in ChREBP-alpha mRNA and protein is not necessarily indicative of increased ChREBP activity. Also, according to the literature, fructose feeding markedly increases the mRNA expression of the ChREBP-beta isoform which is not easily detected by Western Blot. Fructose feeding has variable effect on the ChREBP-alpha isoform, an isoform which is readily detectable on Western Blot, but is not indicative of ChREBP activity. Whether ChREBP is activated in the liver or elsewhere by pregnancy is not clear based on the data provided in this manuscript.
3. Figure 2: Liver ChREBP KO did not effect serum insulin, GTT, or ITT in KPC vs WPC indicating that liver ChREBP has little or no role in normal insulin resistance of pregnancy. Therefore, the motivation and interpretation of experiments studying the role of ChREBP in regulating progesterone or other factors that contribute to insulin resistance of pregnancy is not clear. Moreover, if taken at face value, this would suggest that progesterone, which does decrease in ChREBP KO mice (Fig 3C) does not play a significant role in pregnancy-induced insulin resistance.
4. The authors invoke a complicated mechanism regarding the effects of fructose and ChREBP on progesterone but provide little experimental data to directly investigating this hypothesis. This remains largely speculative.

Other Issues:

5. Fig 1A and B: Few methods are provided regarding these analyses. Panel A and B suggests these are blood samples. Are ChREBP levels in blood relevant to the hypotheses and model proposed in the rest of the paper? In panel B, where are these samples from? Is this a pre-registered study? Is this IRB approved? There is no discussion of this in the methods.

6. Fig 1E: The authors use an anti-ChREBP AB (ABclonal A7630) which has not previously been published and is not commonly used in the field. The representative blot on the vendors website shows a very prominent non-specific band and is not typical of a high-quality blot for ChREBP. While the authors show some evidence that this antibody can detect ChREBP by immunoblot (Panel 7G), the non-specific binding might preclude its use for IHC. Use for IHC should be validated using KO tissues.

7. The authors provide an RRID number regarding the ChREBP floxed mice but search of the RRID database turns up no information for this mouse. More information regarding this floxed mouse, how it was generated, which exons were floxed, how it was validated are required to evaluate this manuscript.

8. Supp Fig 3C. Although there appears to be a strong effect of fructose to affect INSR phosphorylation in pregnancy, there was no clear effect on downstream signaling despite statements suggesting otherwise in the text.

9. Figure 1I: Again, there is no consistent effect of pregnancy (PC vs CC) on the protein levels of ChREBP targets. Also, fructose did not increase the protein levels of ChREBP targets in pregnant mice (PF compared to CF) which suggests that the effects of fructose and pregnancy are not synergistic on hepatic ChREBP activity.

10. Figure 2: Liver ChREBP KO did reduce fructose-induced insulin resistance. However, this is not novel and has previously been published by other groups, though not in the setting of pregnancy. The authors point to an anomalous paper by Jois et al (2017) that suggested that hepatic ChREBP KO impairs glucose homeostasis, but multiple other independent groups have published to the contrary and the results reported by Jois may depend on the nature of the gene deletion performed by these investigators.

Dear reviewer,

Thank you for the review of our manuscript and for your constructive recommendations and comments, which have greatly assisted us in improving the manuscript. We have conducted additional experiments and heavily revised our manuscript. The revised segments of the manuscript are indicated in **YELLOW**. The point-by-point responses to comments are listed below. We anticipate that your comments have been addressed accurately and completely.

Major Issues:

Comment 1: Supplemental Fig2A: There is an increase in the protein levels of some ChREBP targets (Pklr and SCD1) but not others (Fasn, Acly, Elovl6) over the course of pregnancy. The expression of all of these factors can be regulated by additional transcription factors, but the lack of concordance does not strongly support the conclusion that hepatic ChREBP activity increases over the course of pregnancy and undermines a key construct of this manuscript.

Response 1: Thank you for this insightful comment. PKLR is specifically regulated by ChREBP (Haas *et al*, 2012; Liu *et al*, 2017; Yamashita *et al*, 2001). This alignment between PKLR and ChREBP expression gives strong evidence for elevated ChREBP activity throughout pregnancy.

However, the expression of other ChREBP target genes can be impacted by various transcription factors. To further support our findings, we have examined the expression of additional transcription factors, including LXR α and SREBP-1c during pregnancy (as illustrated below), which also could regulate FASN, ACLY, and ELOVL6. Interestingly, both LXR α and SREBP-1c expression levels were reduced in the later stages of pregnancy, as reported by other groups previously (Nikolova *et al*, 2017; Sweeney *et al*, 2006).

Figure for referee with unpublished data and its description has been removed upon request by the authors.

Data Shown to Reviewer: The expression of LXR α and SREBP-1c was reduced during gestation.

(A) Representative Western blot indicating the expression of LXR α and SREBP-1c in pregnant mouse liver.

(B) qRT-PCR results indicating the relative *Srebp-1c* and *Lxra* mRNA levels in the liver of pregnant mice. n = 5, *P<0.05; **P<0.01; ***P<0.001 compared to the E0 group.

This reduction in the expression of LXR α and SREBP-1c aligns with the identified reduction in expression of FASN, ACLY, and ELOVL6 during late pregnancy. Consequently, the concordance in the downregulation of LXR α and SREBP-1c accounts for the decreased expression of Fasn, Acly, and Elovl6 during late pregnancy. According to these facts and your comment, we have removed the results pertaining to FASN, ACLY, and ELOVL6 from the new manuscript to avoid introducing the influences of other transcription factors during pregnancy.

.....
Comment 2: Fig 1D and elsewhere and related to Point 1 above: The authors should measure the expression of distinct ChREBP isoforms by qPCR. Expression of ChREBP-beta tends to be a good index of tissue ChREBP activity whereas expression of ChREBP-alpha both at the gene and protein level is not a strong marker for the ChREBP activity state because its transcriptional activity requires allosteric activation by carbohydrate metabolites. An increase in ChREBP-alpha mRNA and protein is not necessarily indicative of increased ChREBP activity. Also, according to the literature, fructose feeding markedly increases the mRNA

expression of the ChREBP-beta isoform which is not easily detected by Western Blot. Fructose feeding has variable effect on the ChREBP-alpha isoform, an isoform which is readily detectable on Western Blot, but is not indicative of ChREBP activity. Whether ChREBP is activated in the liver or elsewhere by pregnancy is not clear based on the data provided in this manuscript.

Response 2: Thank you for this helpful comment. In response to your suggestions, we have performed additional experiments to examine the expression of ChREBP α and ChREBP β isoforms through qRT-PCR. The findings indicated that the mRNA levels of both ChREBP α and ChREBP β were activated by pregnancy and a high-fructose diet. The results are summarized in the **Figure** below:

Figure for referee with unpublished data and its description has been removed upon request by the authors.

Figure for referee with unpublished data and its description has been removed upon request by the authors.

Figure for referee with unpublished data and its description has been removed upon request by the authors.

Figure for referee with unpublished data and its description has been removed upon request by the authors.

Figure for referee with unpublished data and its description has been removed upon request by the authors.

Figure for referee with unpublished data and its description has been removed upon request by the authors.

Figure for referee with unpublished data and its description has been removed upon request by the authors.

Figure for referee with unpublished data and its description has been removed upon request by the authors.

Figure for referee with unpublished data and its description has been removed upon request by the authors.

Figure for referee with unpublished data and its description has been removed upon request by the authors.

Figure for referee with unpublished data and its description has been removed upon request by the authors.

Figure for referee with unpublished data and its description has been removed upon request by the authors.

Figure for referee with unpublished data and its description has been removed upon request by the authors.

.....
Comment 3: Figure 2: Liver ChREBP KO did not effect serum insulin, GTT, or ITT in KPC vs WPC indicating that liver ChREBP has little or no role in normal insulin resistance of pregnancy. Therefore, the motivation and interpretation of experiments studying the role of

ChREBP in regulating progesterone or other factors that contribute to insulin resistance of pregnancy is not clear. Moreover, if taken at face value, this would suggest that progesterone, which does decrease in ChREBP KO mice (Fig 3C) does not play a significant role in pregnancy-induced insulin resistance.

Response 3: Thank you for this helpful comment. We observed a slight reduction in HOMA-IR, ITT, and GTT in KPC (hepatic ChREBP KO) mice compared to WPC (wild-type) mice under normal-diet conditions, as indicated in the original **Figure 2** (n = 5). However, this trend did not reach significance, which may be due to the limited number of mice used previously. To improve the robustness of our conclusions, we increased the sample size of each group and re-performed the experiments (n = 10). The new results support that hepatic ChREBP deficiency under normal conditions may also contribute to improved insulin sensitivity. However, this does not mean that hepatic ChREBP is bad during pregnancy because the pregnant body requires much more energy from the liver, and proper insulin resistance during pregnancy is necessary.

Figure for referee with unpublished data and its description has been removed upon request by the authors.

The revised data, as outlined in our manuscript, now indicate that liver ChREBP knockout decreases circulating insulin levels, inhibits blood glucose, and ameliorates insulin resistance index (HOMA-IR) and ITT even in the absence of fructose (**NEW Figure 2B-E**). However, GTT did not reach statistical significance (**NEW Figure 2F**). We acknowledge that under normal dietary conditions, the influence of ChREBP on insulin resistance appears relatively weak. It appears that this moderate effect may be advantageous when considering ChREBP as a potential target for insulin level regulation during pregnancy. Maintaining insulin levels within a safe range is critical during pregnancy, and excessive reductions in insulin levels are dangerous.

.....
Comment 4: The authors invoke a complicated mechanism regarding the effects of fructose and ChREBP on progesterone but provide little experimental data to directly investigating this hypothesis. This remains largely speculative.

Response 4: Thank you for this helpful comment. In our investigation, we proposed a mechanism in which a high-fructose diet during pregnancy activated hepatic ChREBP, resulting in increased expression of hepatic MTTP, which, in turn, promotes cholesterol efflux from the liver, increasing progesterone synthesis in the ovaries and placenta.

(1) Fructose-induced ChREBP activation elevates progesterone levels during pregnancy:

We demonstrated that feeding on high-fructose diets during pregnancy results in the activation of hepatic ChREBP at both the transcriptional and protein levels compared to pregnant mice fed a normal diet. Of note, we complemented these findings with RNA-seq analysis data, which showed elevated expression of ChREBP and its target gene in the livers given a high-fructose diet during pregnancy (**NEW Figure 1F**).

Figure for referee with unpublished data and its description has been removed upon request by the authors.

pregnancy; PC group, pregnant mice receiving NC; PF group, pregnant mice receiving HFrD from Day 8 to Day 17 of pregnancy (**E**); RNA-seq assessment of total mouse liver RNA in PF and PC groups. Volcano plot illustrating the transcripts in the liver that were decreased (blue dots), increased (orange dots), or unchanged (gray dots), $n = 3$, $P < 0.05$. (**F**). Expression of *Chrebpa* and *Chrebpb* mRNA was characterized via qRT-PCR, $n = 5$ (**G**); Expression of ChREBP and its downstream targets protein was determined using Western blot in the liver (**H**); * $P < 0.05$, ** $P < 0.01$ compared to the indicated group.

(2) We submit that progesterone can be induced by hepatic ChREBP for the first time:

To provide more evidence, we re-performed experiments involving hepatic ChREBP knockout mice given fructose during the revision period (**NEW Figure 2A**), and the elevated number of samples improved the credibility of our results. We identified that fructose can enhance progesterone levels, and the removal of hepatic ChREBP prevented the increase in progesterone levels (**NEW Figure 3C**). Thus, we consider our conclusion that fructose can promote progesterone through the activation of hepatic ChREBP to be reliable.

Figure for referee with unpublished data and its description has been removed upon request by the authors.

Figure for referee with unpublished data and its description has been removed upon request by the authors.

Figure for referee with unpublished data and its description has been removed upon request by the authors.

Figure for referee with unpublished data and its description has been removed upon request by the authors.

Figure for referee with unpublished data and its description has been removed upon request by the authors.

Figure for referee with unpublished data and its description has been removed upon request by the authors.

.....
Other Issues:

Comment 5: Fig 1A and B: Few methods are provided regarding these analyses. Panel A and B suggests these are blood samples. Are ChREBP levels in blood relevant to the hypotheses and model proposed in the rest of the paper? In panel B, where are these samples from? Is this a pre-registered study? Is this IRB approved? There is no discussion of this in the methods.

Response 5: Thank you very much for this helpful comment and questions.

(1) Analysis methods for Figure 1A and B:

The method in **Figure 1A** (moved to **NEW Supplementary Figure 13A** in the revised manuscript) has been written in the Research Design and Methods section in Supplemental Data. (*Line 238-245 of Page 23 in supplemental data*). We have also provided a comprehensive description of the methods employed for **Figure 1B** (moved to **NEW Figure 1I** in the revised manuscript) in the Research Design and Methods section, including sample collection, processing, and ChREBP mRNA quantification. (*Line 552-557 of Page 26 in manuscript*)

(2) Relevance of blood samples:

Our primary focus is on ChREBP levels in the liver. Your feedback prompted us to reconsider the presentation of our data. We acknowledge that the use of human blood samples in **Figure 1A and B** may not be directly aligned with the central concentration of our paper. In light of this, we have relocated the data from human blood samples in **Figure 1A and B** to **NEW Supplementary Figure 13A and Figure 1I**.

Given the challenge of acquiring liver samples during pregnancy, we aimed to assess whether ChREBP levels in more easily accessible blood samples could be a substitute marker for ChREBP activity throughout pregnancy. We investigated the GEO database and determined that the transcriptional level of ChREBP was significantly elevated in peripheral blood mononuclear cells of donors at the 9th month (late stage of pregnancy) of pregnancy compared to non-pregnant subjects (**NEW Supplementary Figure 13A**). In mice blood, we also identified that the transcriptional level of ChREBP was significantly elevated in donors at the late stage of pregnancy compared to non-pregnant mice (**NEW Supplementary Figure 13B**). Additionally, ChREBP mRNA levels in the liver are increased during mouse pregnancy (**NEW Figure 1B**), suggesting that whole-blood ChREBP may be an indicator of the enhanced ChREBP expression in the liver throughout pregnancy. To support this concept, compared to healthy pregnant women, ChREBP mRNA levels also increased in human GDM patients (**NEW Figure 1I**). In mouse blood samples, we observed an increase in ChREBP mRNA levels following the administration of an HFrD diet (**NEW Supplementary Figure 13C**). In addition, ChREBP mRNA levels in the liver increased during mice pregnancy with HFrD-diet (**NEW Figure 1H**). These results demonstrated that blood ChREBP levels are consistent with hepatic ChREBP expression. Therefore, the mRNA levels of ChREBP in blood samples may be an alternative marker for hepatic ChREBP activity during pregnancy. We have acknowledged the limitations of these findings in the "Discussion" section of our

paper to ensure full transparency and clarity. (*Line 406-416 of Page 19 and Line 417-427 of Page 20 in manuscript*)

(3) Sample origin and ethical considerations:

Figure 1B (moved to **NEW Figure II** in the revised manuscript) includes blood samples from both normal pregnant women and pregnant women with gestational diabetes. We affirm that this research was conducted in accordance with ethical guidelines, and have obtained the necessary Institutional Review Board (IRB) approval. We will include this information in the Methods section of the manuscript to give transparency and address your concerns. (*Line 567-569 of Page 26 in manuscript*). Additionally, we will submit the IRB approval documentation for your reference, attached at the end of the *Response to Reviewers* file.

.....
Comment 6: Fig 1E: The authors use an anti-ChREBP AB (ABclonal A7630) which has not previously been published and is not commonly used in the field. The representative blot on the vendor's website shows a very prominent non-specific band and is not typical of a high-quality blot for ChREBP. While the authors show some evidence that this antibody can detect ChREBP by immunoblot (Panel 7G), the non-specific binding might preclude its use for IHC. Use for IHC should be validated using KO tissues.

Response 6: Thank you for this insightful comment. We employed the Novus NB400-135 antibody for our immunofluorescence experiments in the original manuscript, including **Figure 1E** (**NEW Figure 1C** in the revised manuscript). We included the Novus NB400-135 antibody in the supplementary data section of the original manuscript's antibody list (the table containing the Novus antibody in the original manuscript is outlined in the table below). However, we did not specify the exact experiments for which each antibody was used. We have now indicated this in the supplementary data section. (*Line 30-31 of Page 3 in supplemental data*)

Figure for referee with unpublished data and its description has been removed upon request by the authors.

Data Shown to Reviewer: The table of reagents utilized in the experiments in the original manuscript.

The Novus NB400-135 antibody has been confirmed for use in immunohistochemistry experiments (https://www.novusbio.com/products/chrebp-antibody_nb400-135#datasheet). Additionally, we performed immunohistochemistry experiments on mouse liver paraffin sections utilizing the Novus NB400-135 antibody, as illustrated in the **Figure** below.

Figure for referee with unpublished data and its description has been removed upon request by the authors.

Data Shown to Reviewer: Immunohistochemistry of ChREBP (Novus NB400-135) in mouse liver.

Regarding the ABclonal A7630 antibody, we used it for Western blot experiments, as previously used and validated for this purpose in published papers (Pu *et al*, 2021; Qiao *et al*, 2022; Zhao *et al*, 2022; Zhou *et al*, 2023). We conducted knockout experiments in which we employed livers from h*Chrebp*-KO mice. The absence of ChREBP protein in these samples acted as a negative control, enabling us to confirm that the ABclonal A7630 antibody recognizes ChREBP (**NEW Supplemental Figure 4D**).

Figure for referee with unpublished data and its description has been removed upon request by the authors.

Figure for referee with unpublished data and its description has been removed upon request by the authors.

Data Shown to Reviewer: Expression of the ChREBP protein in the liver was determined using the Novus NB400-135 antibody.

.....
Comment 7: The authors provide an RRID number regarding the ChREBP floxed mice but search of the RRID database turns up no information for this mouse. More information regarding this floxed mouse, how it was generated, which exons were floxed, how it was validated are required to evaluate this manuscript.

Response 7: Thank you for this comment. The ChREBP floxed mice employed in our study were constructed by GemPharmatech (Jiangsu, China, <https://cn.gempharmatech.com/>), a commercial company specializing in the development of genetically modified mice.

(1) Overview of the strategy used to generate the ChREBP floxed mice:

The *Mlxipl* gene has 10 transcripts. According to the *Mlxipl* gene structure, exon 2 of the *Mlxipl*-201 (ENSMUST00000005507.9) transcript was recommended as the knockout region. The region contains a 107 bp coding sequence, and knocking out the region will result in the disruption of protein function.

Figure for referee with unpublished data and its description has been removed upon request by the authors.

Data Shown to Reviewer: *Mlxipl* genomic location distribution.

(2) In this project, we utilized CRISPR/Cas9 to modify the *Mlxipl* gene:

The brief process is as follows: sgRNA was transcribed *in vitro*, and the donor vector was constructed. Cas9, sgRNA, and Donor were microinjected into the fertilized eggs of C57BL/6J mice. Fertilized eggs were transplanted to obtain positive F0 mice, which were confirmed using PCR and sequencing. A stable F1 generation mouse model was acquired by mating positive F0 generation mice with C57BL/6J mice.

Figure for referee with unpublished data and its description has been removed upon request by the authors.

Data Shown to Reviewer: Model using CRISPR/Cas9 technology to edit the *Mlxipl* gene.

(3) *The reproductive strategies of hepatocyte-specific ChREBP deficient mice:*

The floxed mice had genes knocked out following mating with mice expressing Cre recombinase, resulting in the loss of function for the target gene in hepatocytes.

Figure for referee with unpublished data and its description has been removed upon request by the authors.

Data Shown to Reviewer: Strategy for Genotyping and Primer Information.

Figure for referee with unpublished data and its description has been removed upon request by the authors.

Data Shown to Reviewer: Gel Image (provided by GemPharmatech).

Figure for referee with unpublished data and its description has been removed upon request by the authors.

Data Shown to Reviewer: Representative gel photographs illustrating the PCR products amplified from the mouse tail to identify the genotype in this study.

.....
Comment 8: Supp Fig 3C. Although there appears to be a strong effect of fructose to affect INSR phosphorylation in pregnancy, there was no clear effect on downstream signaling despite statements suggesting otherwise in the text.

Response 8: Thank you for this insightful comment. Upon evaluating our data and reviewing the grayscale analysis in **Supplementary Figure 3C (NEW Supplementary Figure 3D)**, we determined that the original bands did indicate an impact of fructose on downstream INSR signaling p-AKT and p-GSK3 β . However, we acknowledge that the original bands were not as visually clear as they should have been, which may have hampered accurate assessment of the trend. To rectify this issue and ensure the accuracy of our findings, we repeated the experiment and acquired new, more discernible bands for downstream signaling. These revised results give a clearer and more precise representation of the effects of fructose on INSR downstream signaling throughout pregnancy.

Figure for referee with unpublished data and its description has been removed upon request by the authors.

.....
Comment 9: Figure 1I: Again, there is no consistent effect of pregnancy (PC vs CC) on the protein levels of ChREBP targets. Also, fructose did not increase the protein levels of ChREBP targets in pregnant mice (PF compared to CF) which suggests that the effects of fructose and pregnancy are not synergistic on hepatic ChREBP activity.

Response 9: Thank you for this helpful comment.

(1) *Inconsistent influence of pregnancy on ChREBP targets:*

We acknowledge that there is an inconsistent influence of pregnancy on the protein levels of ChREBP targets in **Figure 1I** (moved to **NEW Figure 1H** in the revised manuscript). This inconsistency arises from the fact that the PC group represents late-pregnancy mice, and compared to non-pregnant conditions (CC), only PKLR and SCD1 exhibit upregulation, while other ChREBP targets exhibit downregulation. This observation is consistent with the findings presented in **Figure S2A**. We have outlined the specific explanation in Response 1. Briefly, we believe that this variability in ChREBP target expression during late pregnancy may be impacted by the activity of other transcription factors, namely LXR and SREBP1c, which can tune the expression of these targets.

(2) Synergistic effects of fructose and pregnancy on hepatic ChREBP activity:

The CF group represents non-pregnant mice administered a fructose diet, while the PC group represents pregnant mice administered a regular diet. To examine whether fructose increases the expression of ChREBP targets in pregnant mice, we compared the PF (pregnant mice fed fructose) group to the PC group. Upon grayscale analysis, we identified that the bands did indeed indicate an increase in the PF group compared to the PC group with respect to ChREBP. (*Line 175 of Page 18 in supplemental data*)

According to your suggestion, we examined the mRNA levels of *Chrebbp* (**NEW Figure 3G**). The findings indicated a significant elevation in *Chrebbp* levels in the PF group compared to the PC groups.

Figure for referee with unpublished data and its description has been removed upon request by the authors.

Figure for referee with unpublished data and its description has been removed upon request by the authors.

Data Shown to Reviewer: Expression of ChREBP protein in the liver determined using the Novus NB400-135 antibody.

Therefore, fructose and pregnancy exhibit synergistic effects on hepatic ChREBP activity.

.....
Comment 10: Figure 2: Liver ChREBP KO did reduce fructose-induced insulin resistance. However, this is not novel and has previously been published by other groups, though not in the setting of pregnancy. The authors point to an anomalous paper by Jois et al (2017) that suggested that hepatic ChREBP KO impairs glucose homeostasis, but multiple other independent groups have published to the contrary and the results reported by Jois may depend on the nature of the gene deletion performed by these investigators.

Response 10: Thank you for this comment. In **Figure 2**, we performed experiments by feeding either h*Chrebp*-KO mice or *Chrebp*^{flox/flox} mice a high-fructose or normal diet during pregnancy. Subsequently, we examined their glucose tolerance (GTT), insulin tolerance (ITT), blood glucose, and insulin levels in the late stages of pregnancy. The findings indicate an increase in insulin sensitivity in pregnant mice with hepatic ChREBP knockout compared to control mice.

Figure for referee with unpublished data and its description has been removed upon request by the authors.

Figure for referee with unpublished data and its description has been removed upon request by the authors.

Data Shown to Reviewer: Summary of liver-specific ChREBP knockout roles in experimental mice models.

It is critical to note that none of these studies specifically assessed the relationship between hepatic ChREBP and insulin resistance **in females**. Acknowledging that there are notable differences between male and female mice in growth, metabolism, hormone levels, and other physiological features, we focused on understanding how ChREBP functions in insulin resistance, specifically in females. Our findings indicate that the impact of maternal high-fructose feeding on offspring has gender-specific differences associated with ChREBP (**NEW Figure 6 and Figure 7**). This underscores the importance of studying ChREBP in a sex-specific context.

Pregnancy is linked to unique physiological changes, such as increased lipid accumulation, and hormonal alterations. These characteristics suggest that ChREBP's role in pregnancy may differ from its role in non-pregnant states. As Katz and colleagues have determined in their review, the impacts of hepatic ChREBP on insulin sensitivity and glucose homeostasis vary depending on specific physiological or pathological conditions or nutritional contexts (Katz *et al*, 2021). Our study determined that ChREBP's regulation of insulin resistance in pregnancy is associated with progesterone, significantly increasing during gestation. This finding adds a novel dimension to the understanding of ChREBP's role and highlights its significance in the context of pregnancy.

As outlined in the above table, the knockout sites employed in different studies are inconsistent. The outcomes reported by Jois may depend on the nature of the gene deletions performed. However, since the feeding methods for mice are variable, we believe that nutritional status could also be a contributing factor leading to contradictory results. Our study proposes that both a high-fructose diet and pregnancy status can encourage insulin resistance through the ChREBP/progesterone pathway. Previous research has demonstrated that a high-fructose diet and pregnancy promote progesterone receptor expression, but a high-fat diet inhibits progesterone receptor expression (Ahn *et al*, 2022; Coyoy-Salgado *et al*, 2020; Heard *et al*, 2016). Therefore, we provide an alternative perspective to characterize the current contradiction: under high-fructose or pregnancy conditions, the ChREBP/progesterone/insulin resistance signaling pathway is activated, and knocking out ChREBP significantly inhibits insulin resistance. However, under high-fat conditions, ChREBP/progesterone may be continuously inhibited, and knocking out ChREBP may regulate insulin sensitivity via other pathways (such as modulating autocrine signals).

In light of these considerations, while studies on the relationship between liver ChREBP and insulin resistance have been conducted, our investigation is unique and necessary due to the specific focus on females and pregnancy.

References:

- Ahn SH, Nguyen SL, Kim TH, Jeong JW, Arora R, Lydon JP, Petroff MG (2022) Nuclear Progesterone Receptor Expressed by the Cortical Thymic Epithelial Cells Dictates Thymus Involution in Murine Pregnancy. *Frontiers in endocrinology* 13: 846226
- Arentson-Lantz EJ, Zou M, Teegarden D, Buhman KK, Donkin SS (2016) Maternal high fructose and low protein consumption during pregnancy and lactation share some but not all effects on early-life growth and metabolic programming of rat offspring. *Nutrition research (New York, NY)* 36: 937-946
- Coyoy-Salgado A, Segura-Uribe JJ, Manuel Gallardo J, Estrada-Cruz NA, Camacho-Arroyo I, Guerra-Araiza C (2020) Tibolone regulates systemic metabolism and the expression of sex hormone receptors in the central nervous system of ovariectomised rats fed with high-fat and high-fructose diet. *Brain research* 1748: 147096
- Haas JT, Miao J, Chanda D, Wang Y, Zhao E, Haas ME, Hirschey M, Vaitheesvaran B, Farese RV, Jr., Kurland IJ, Graham M, Crooke R, Foufelle F, Biddinger SB (2012) Hepatic insulin signaling is required for obesity-dependent expression of SREBP-1c mRNA but not for feeding-dependent expression. *Cell metabolism* 15: 873-84
- Heard ME, Melnyk SB, Simmen FA, Yang Y, Pabona JM, Simmen RC (2016) High-Fat Diet Promotion of Endometriosis in an Immunocompetent Mouse Model is Associated With Altered Peripheral and Ectopic Lesion Redox and Inflammatory Status. *Endocrinology* 157: 2870-82
- Katz LS, Baumel-Alterzon S, Scott DK, Herman MA (2021) Adaptive and maladaptive roles for ChREBP in the liver and pancreatic islets. *The Journal of biological chemistry* 296: 100623
- Liu G, Zhou L, Zhang H, Chen R, Zhang Y, Li L, Lu JY, Jiang H, Liu D, Qi S, Jiang YM, Yin K, Xie Z, Shi Y, Liu Y, Cao X, Chen YX, Zou D, Zhang WJ (2017) Regulation of hepatic lipogenesis by the zinc finger protein Zbtb20. *Nature communications* 8: 14824
- Mason TM (1998) The role of factors that regulate the synthesis and secretion of very-low-density lipoprotein by hepatocytes. *Critical reviews in clinical laboratory sciences* 35: 461-87
- Nikolova V, Papacleovoulou G, Bellafante E, Borges Manna L, Jansen E, Baron S, Abu-Hayyeh S, Parker M, Williamson C (2017) Changes in LXR signaling influence early-pregnancy lipogenesis and protect against dysregulated fetoplacental lipid homeostasis. *American journal of physiology Endocrinology and metabolism* 313: E463-e472
- Pu Y, Xiang J, Zhang X, Deng Y, Liu H, Tan W (2021) CD36 as a Molecular Target of Functional DNA Aptamer NAFLD01 Selected against NAFLD Cells. *Analytical chemistry* 93: 3951-3958
- Qiao P, Jia Y, Ma A, He J, Shao C, Li X, Wang S, Yang B, Zhou H (2022) Dapagliflozin protects against nonalcoholic steatohepatitis in db/db mice. *Frontiers in pharmacology* 13: 934136
- Sweeney TR, Moser AH, Shigenaga JK, Grunfeld C, Feingold KR (2006) Decreased nuclear hormone receptor expression in the livers of mice in late pregnancy. *American journal of physiology Endocrinology and metabolism* 290: E1313-20
- Yamashita H, Takenoshita M, Sakurai M, Bruick RK, Henzel WJ, Shillinglaw W, Arnot D, Uyeda K (2001) A glucose-responsive transcription factor that regulates carbohydrate

metabolism in the liver. *Proceedings of the National Academy of Sciences of the United States of America* 98: 9116-21

Zhao F, Zhang L, Zhang M, Huang J, Zhang J, Chang Y (2022) FGF9 Alleviates the Fatty Liver Phenotype by Regulating Hepatic Lipid Metabolism. *Frontiers in pharmacology* 13: 850128

Zhou X, Zhang X, Niu D, Zhang S, Wang H, Zhang X, Nan F, Jiang S, Wang B (2023) Gut microbiota induces hepatic steatosis by modulating the T cells balance in high fructose diet mice. *Scientific reports* 13: 6701

Figure for referee with unpublished data and its description has been removed upon request by the authors.

Figure for referee with unpublished data and its description has been removed upon request by the authors.

Figure for referee with unpublished data and its description has been removed upon request by the authors.

Dear Prof. Duan,

Thank you for submitting your revised manuscript. It has now been seen by all of the original referees.

My apologies for the delay in getting back to you - it took longer than anticipated to receive the referee reports given this busy time of the year.

As you can see, the referees find that the study is significantly improved during revision and recommend publication. However, I need you to address the points below before I can accept the manuscript.

- Please address the remaining concerns of referees #3 as below and provide a point-by-point response. Also, please highlight the changes in the text.
- Comment 1 and 2: Please discuss the caveats pointed out by the referee in the manuscript text.
- Comment 3: Please perform the textual changes requested by the referee.
- Comment 4: Please either test the proposed role of cholesterol delivery experimentally or revise the text as suggested by the referee.
- Comment 7: Please perform the validation experiments suggested by the referee and include the details on the floxed mice into the manuscript.

Please let me know if you would like to discuss any of the points further.

- We can accommodate maximum 5 keywords due to technical reasons. Therefore, please remove 2 of the keywords.
- Please remove the Author Contributions section from the manuscript.
- As per our format requirements, in the reference list, citations should be listed in alphabetical order and then chronologically, with the authors' surnames and initials inverted; where there are more than 10 authors on a paper, 10 will be listed, followed by 'et al.'. Please see <https://www.embopress.org/page/journal/14693178/authorguide#referencesformat>
- We note that Dr. Jihong Han's ORCID iD is currently not linked. EMBO Press policy asks for all corresponding authors to link to their ORCID iDs. You can read about the change under "Authorship Guidelines" in the Guide to Authors here: <https://www.embopress.org/page/journal/14693178/authorguide#authorshipguidelines>

In order to link your ORCID iD to your account in our manuscript tracking system, please do the following:

1. Click the 'Modify Profile' link at the bottom of your homepage in our system.
2. On the next page you will see a box halfway down the page titled ORCID*. Below this box is red text reading 'To Register/Link to ORCID, click here'. Please follow that link: you will be taken to ORCID where you can log in to your account (or create an account if you don't have one)
3. You will then be asked to authorise Wiley to access your ORCID information. Once you have approved the linking, you will be brought back to our manuscript system.

We regret that we cannot do this linking on your behalf for security reasons.

- Please resubmit Source Data as one file per figure. Source data files need to be submitted as zipped folders, one .zip file for each figure. Inside each folder, the files should be organized in subfolders, one subfolder for each panel.
- In the Appendix File, we note that Table S1 and S2 are reagents that are used in the study. Therefore, please submit these tables as Structured Methods Tables in word format (please see for template <https://www.embopress.org/page/journal/14693178/authorguide#structuredmethods>).
- We note that the Appendix Table currently contains 13 Supplemental Figures. I would encourage you to take advantage of the Expanded View figure type by converting maximum 5 of the figures into Expanded View Figures, in which case they should be called out as Figure EV1 etc. and their legends need to be moved to the main manuscript text after the main figure legends (please see <https://www.embopress.org/page/journal/14693178/authorguide#expandedview> for further info). The rest of the figures would then remain in the Appendix. Please rename the Appendix figures as Appendix Figure S1 etc and update the callouts accordingly.
- We note that the tables in the Appendix file (page 18-22) are the results of statistical analysis of band density for Western blots of the figures. Please remove these tables from the Appendix file and submit them as source data.
- Please move the text on the pages 23 and 24 of the Appendix file to the Materials & Methods section of the manuscript text, as no additional materials and methods are permitted outside of the main manuscript text. Methods are excluded from the word count.
- Guarantor statement needs to be removed from the manuscript text.
- In the figure legends, please move the definitions of the annotated p values ***/**/* to the Data Information section of all figures (please see <https://www.embopress.org/page/journal/14693178/authorguide#conventionsabbreviations> for examples).
- Please note that the figure legend for figures 7f-g is not provided in a sequential manner (i.e., legend for figure 7g is provided before legend 7f). This needs to be rectified.
- Please note that the asterisk "****" is not defined in the legend of figure 5c. This needs to be rectified.
- Please make the datasets GSE247892, GSE248051, GSE247892 and GSE248051 publicly available and remove the reviewer

token=s from the manuscript.

- Our production/data editors have asked you to clarify several points in the figure legends:
 - Please note that a separate 'Data Information' section is required in the legends of all the figures. (please see <https://www.embopress.org/page/journal/14693178/authorguide#figureformat> for examples)
 - Please indicate the statistical test used for data analysis in the legends of figures 4a
- Papers published in EMBO Reports include a 'synopsis' and 'bullet points' to further enhance discoverability. Both are displayed on the html version of the paper and are freely accessible to all readers. The synopsis includes a short standfirst summarizing the study in 1 or 2 sentences (max 35 words) that summarize the paper and are provided by the authors and streamlined by the handling editor. I would therefore ask you to include your synopsis blurb and 3-5 bullet points listing the key experimental findings.
- In addition, please provide an image for the synopsis. This image should provide a rapid overview of the question addressed in the study but still needs to be kept fairly modest since the image size cannot exceed 550 (width) x 300-600 (height) pixels.

Thank you again for giving us to consider your manuscript for EMBO Reports, I look forward to your minor revision.

Kind regards,

Deniz Senyilmaz Tiebe

--

Deniz Senyilmaz Tiebe, PhD
Editor
EMBO Reports

Referee #1:

The authors have largely improved the quality of the manuscript and replied to all reviewer's concerns.

Referee #2:

I am really satisfied with the revised version of the manuscript. The authors have covered full my comments and the revised version of the manuscript is substantially improved and more clear compared to the previous version. I am recommending publication in its current form.

Referee #3:

The authors have made significant efforts to address my prior concerns including numerous new experiments. The manuscript is markedly improved and contains many new and interesting observations worthy of publication. However, some questions remain inadequately addressed.

Comment 1 and 2: The previously expressed concerns regarding whether liver ChREBP activity is increased during pregnancy are largely mitigated by the authors' response to Comment 2 and associated new data throughout the manuscript indicating that ChREBP-beta mRNA is modestly increased in the liver over the course of pregnancy and other relevant conditions tested in this manuscript. However, the authors should carefully consider editing some of the other text. In my initial review, I noted that as shown in original Supplemental Fig2A, there is an increase in the protein levels of some ChREBP targets (Pklr and SCD1) but not others (Fasn, Acly, Elovl6) over the course of pregnancy. I noted that the expression of all of these factors are known ChREBP targets, but each of them can also be regulated by additional transcription factors. The lack of a concordant increase across these known ChREBP targets seems to conflict with the conclusion that hepatic ChREBP activity increases over the course of pregnancy and undermines a key construct of this manuscript. The authors 'argue' that Pklr is specifically regulated by ChREBP but that the expression of other targets can be impacted by other transcription factors. This is not accurate as Pklr can also be regulated by other transcription factors - for instance c-Myc, HNF1a, and HNF4a can all regulate Pklr (PMID: 20382893). Myc can also regulate Scd1 (PMID: 28474697). The authors provide some additional data (for reviewer only) to suggest that downregulation of Srebp1c and LXRA during pregnancy might account for the lack of induction of lipogenic enzymes like Fasn, Acly, and Elovl6 that are also known ChREBP targets. These blots are of moderate quality and not very convincing. Moreover, only the immature 100 kd form of Srebp1c is shown which does not reflect Srebp1c activity. Likewise, LXRA is a ligand dependent TF and its protein level is not a good indicator of its activity. Moreover, Scd1 is both a Srebp1c and LXRA target yet increases in the liver during pregnancy. The authors should not rely on this data or these arguments and this may require some revision to the text.

Comment 3: The authors have performed new experiments in Figure 2 with larger numbers of mice per group to assess the impact of liver ChREBP on insulin resistance of pregnancy in both normal dietary conditions and with high-fructose feeding. The effects of liver ChREBP KO to reduce insulin resistance in the fructose fed state are robust. The original experiment showed

minimal or no effect of liver ChREBP KO in on glycemic indices in the chow fed condition. The new experiments do show a statistically significant effect of ChREBP KO to reduce enhance some, but not all, indices of insulin sensitivity though the effect sizes are small. The authors acknowledge that under normal dietary conditions, the influence of hepatic ChREBP on insulin resistance of pregnancy is weak. The authors do not address the second part of this comment. The (new) modest improvements in insulin sensitivity with liver ChREBP KO in normal dietary conditions do seem to parallel a modest reduction in progesterone with liver ChREBP KO (Fig 3C) suggesting that in normal dietary conditions, ChREBP has only a modest effect on progesterone production and the insulin resistance of pregnancy. Again, overall, these results indicate that in normal dietary conditions, liver ChREBP has marginal impact on these important aspects of pregnancy. This should be clear in the abstract and throughout the rest of the manuscript.

Comment 4: The authors propose that fructose feeding induced hepatic ChREBP/MTTP which increases circulating cholesterol and that this drives increased progesterone synthesis exacerbating pregnancy associated insulin resistance. The authors provide new and convincing evidence that fructose feeding can induce progesterone and that this induction is ChREBP dependent (New Figure 3C). This is interesting and exciting. However, the mechanism by which this occurs remains speculative. The authors propose that the ability of fructose to activate ChREBP and induce MTTP enhances cholesterol efflux and that this drives progesterone synthesis by providing more cholesterol substrate for its synthesis. However, there is extensive literature examining biosynthesis of sex hormones and it does not support the idea that cholesterol delivery is limiting for sex steroid synthesis or that increasing circulating cholesterol can drive an increase sex steroid hormone synthesis. As one example, many clinical studies show that statin used which has a large effect to reduce circulating cholesterol in people, has no impact on circulating testosterone in men or estradiol in women. This is likely because the amount of circulating cholesterol FAR exceeds the amount required for sex steroid hormone synthesis by many orders of magnitude. The authors provide evidence that circulating levels of cholesterol do increase with fructose feeding and decline with ChREBP KO, but there is abundant circulating cholesterol in all conditions (New Fig 4B) and again, this is in far excess of the amount of cholesterol that hormone synthetic tissues require to synthesize sex steroid hormones. The authors do provide very interesting data that MTTP overexpression in liver ChREBP KO mice does increase circulating progesterone. However, it is more likely through some indirect mechanism mediated by lipid droplets or other effects on the liver than by increasing the delivery of cholesterol to be used as substrate. The authors could test their cholesterol delivery hypothesis directly or markedly revise the text around these claims.

Comment 7: The authors provided data regarding their gene targeting strategy to generate the ChREBP floxed mice used in this manuscript. This is a new model not previously published. Details of the strategy need to be included in the manuscript including targeted exons, at least in supplementary data. The authors have target exon 2 which potentially leaves the alternative translation start site documented to exist in exon 4 intact. The authors should perform additional validation studies to show that a wide range of canonical ChREBP transcriptional targets are downregulated at the mRNA level the in liver ChREBP KO. This should also be included in supplementary data.

Response to the comments from the Reviewer #3

Referee #3:

The authors have made significant efforts to address my prior concerns including numerous new experiments. The manuscript is markedly improved and contains many new and interesting observations worthy of publication. However, some questions remain inadequately addressed.

Comment 1 and 2: The previously expressed concerns regarding whether liver ChREBP activity is increased during pregnancy are largely mitigated by the authors' response to Comment 2 and associated new data throughout the manuscript indicating that ChREBP-beta mRNA is modestly increased in the liver over the course of pregnancy and other relevant conditions tested in this manuscript. However, the authors should carefully consider editing some of the other text. In my initial review, I noted that as shown in original Supplemental Fig2A, there is an increase in the protein levels of some ChREBP targets (Pklr and SCD1) but not others (Fasn, Acly, Elovl6) over the course of pregnancy. I noted that the expression of all of these factors are known ChREBP targets, but each of them can also be regulated by additional transcription factors. The lack of a concordant increase across these known ChREBP targets seems to conflict with the conclusion that hepatic ChREBP activity increases over the course of pregnancy and undermines a key construct of this manuscript. The authors 'argue' that Pklr is specifically regulated by ChREBP but that the expression of other targets can be impacted by other transcription factors. This is not accurate as Pklr can also be regulated by other transcription factors - for instance c-Myc, HNF1a, and HNF4a can all regulate Pklr (PMID: 20382893). Myc can also regulate Scd1 (PMID: 28474697). The authors provide some additional data (for reviewer only) to suggest that downregulation of Srebp1c and LXRA during pregnancy might account for the lack of induction of lipogenic enzymes like Fasn, Acly, and Elovl6 that are also known ChREBP targets. These blots are of moderate quality and not very convincing. Moreover, only the immature 100 kd form of Srebp1c is shown which does not reflect Srebp1c activity. Likewise, LXRA is a ligand dependent TF and its protein level is not a good indicator of its activity. Moreover, Scd1 is both a Srebp1c and LXRA target yet increases in the liver during pregnancy. The authors should not rely on this data or these arguments and this may require some revision to the text.

Comment 3: The authors have performed new experiments in Figure 2 with larger numbers of mice per group to assess the impact of liver ChREBP on insulin resistance of pregnancy in both normal dietary conditions and with high-fructose feeding. The effects of liver ChREBP KO to reduce insulin resistance in the fructose fed state are robust. The original experiment showed minimal or no effect of liver ChREBP KO in on glycemic indices in the chow fed condition. The new experiments do show a statistically significant effect of ChREBP KO to reduce enhance some, but not all, indices of insulin sensitivity though the effect sizes are small. The authors acknowledge that under normal dietary conditions, the influence of hepatic ChREBP on insulin resistance of pregnancy is weak. The authors do not address the second part of this comment. The (new) modest improvements in insulin sensitivity with liver ChREBP KO in normal dietary conditions do seem to parallel a modest reduction in progesterone with liver ChREBP KO (Fig 3C) suggesting that in normal dietary conditions,

ChREBP has only a modest effect on progesterone production and the insulin resistance of pregnancy. Again, overall, these results indicate that in normal dietary conditions, liver ChREBP has marginal impact on these important aspects of pregnancy. This should be clear in the abstract and throughout the rest of the manuscript.

Comment 4: The authors propose that fructose feeding induced hepatic ChREBP/MTTP which increases circulating cholesterol and that this drives increased progesterone synthesis exacerbating pregnancy associated insulin resistance. The authors provide new and convincing evidence that fructose feeding can induce progesterone and that this induction is ChREBP dependent (New Figure 3C). This is interesting and exciting. However, the mechanism by which this occurs remains speculative. The authors propose that the ability of fructose to activate ChREBP and induce MTTP enhances cholesterol efflux and that this drives progesterone synthesis by providing more cholesterol substrate for its synthesis. However, there is extensive literature examining biosynthesis of sex hormones and it does not support the idea that cholesterol delivery is limiting for sex steroid synthesis or that increasing circulating cholesterol can drive an increase sex steroid hormone synthesis. As one example, many clinical studies show that statin used which has a large effect to reduce circulating cholesterol in people, has no impact on circulating testosterone in men or estradiol in women. This is likely because the amount of circulating cholesterol FAR exceeds the amount required for sex steroid hormone synthesis by many orders of magnitude. The authors provide evidence that circulating levels of cholesterol do increase with fructose feeding and decline with ChREBP KO, but there is abundant circulating cholesterol in all conditions (New Fig 4B) and again, this is in far excess of the amount of cholesterol that hormone synthetic tissues require to synthesize sex steroid hormones. The authors do provide very interesting data that MTTP overexpression in liver ChREBP KO mice does increase circulating progesterone. However, it is more likely through some indirect mechanism mediated by lipid droplets or other effects on the liver than by increasing the delivery of cholesterol to be used as substrate. The authors could test their cholesterol delivery hypothesis directly or markedly revise the text around these claims.

Comment 7: The authors provided data regarding their gene targeting strategy to generate the ChREBP floxed mice used in this manuscript. This is a new model not previously published. Details of the strategy need to be included in the manuscript including targeted exons, at least in supplementary data. The authors have target exon 2 which potentially leaves the alternative translation start site documented to exist in exon 4 intact. The authors should perform additional validation studies to show that a wide range of canonical ChREBP transcriptional targets are downregulated at the mRNA level the in liver ChREBP KO. This should also be included in supplementary data.

Dear reviewer,

Thank you for the review of our manuscript and for your constructive recommendations and comments, which have greatly assisted us in improving quality of the manuscript. Accordingly, we have conducted additional experiments and heavily revised our manuscript. The revised segments of the manuscript are indicated in **YELLOW**. The point-by-point

responses to comments are listed below. We anticipate that your comments have been addressed accurately and completely.

Comment 1 and 2: The previously expressed concerns regarding whether liver ChREBP activity is increased during pregnancy are largely mitigated by the authors' response to Comment 2 and associated new data throughout the manuscript indicating that ChREBP-beta mRNA is modestly increased in the liver over the course of pregnancy and other relevant conditions tested in this manuscript. However, the authors should carefully consider editing some of the other text. In my initial review, I noted that as shown in original Supplemental Fig2A, there is an increase in the protein levels of some ChREBP targets (Pklr and SCD1) but not others (Fasn, Acly, Elovl6) over the course of pregnancy. I noted that the expression of all of these factors are known ChREBP targets, but each of them can also be regulated by additional transcription factors. The lack of a concordant increase across these known ChREBP targets seems to conflict with the conclusion that hepatic ChREBP activity increases over the course of pregnancy and undermines a key construct of this manuscript. The authors 'argue' that Pklr is specifically regulated by ChREBP but that the expression of other targets can be impacted by other transcription factors. This is not accurate as Pklr can also be regulated by other transcription factors - for instance c-Myc, HNF1a, and HNF4a can all regulate Pklr (PMID: 20382893). Myc can also regulate Scd1 (PMID: 28474697). The authors provide some additional data (for reviewer only) to suggest that downregulation of Srebp1c and LXRA during pregnancy might account for the lack of induction of lipogenic enzymes like Fasn, Acly, and Elovl6 that are also known ChREBP targets. These blots are of moderate quality and not very convincing. Moreover, only the immature 100 kd form of Srebp1c is shown which does not reflect Srebp1c activity. Likewise, LXRA is a ligand dependent TF and its protein level is not a good indicator of its activity. Moreover, Scd1 is both a Srebp1c and LXRA target yet increases in the liver during pregnancy. The authors should not rely on this data or these arguments and this may require some revision to the text.

Response 1 and 2: Thank you for the helpful comments. We recognize that PKLR is not a specific target gene of ChREBP, and we have already avoided using this term in the text. Regarding the inconsistent expression levels of the target genes of ChREBP, we are not yet clear about the specific mechanisms involved. Given the multitude of molecules involved in regulating lipid synthesis during pregnancy, we believe that there may be other transcription factors or post-transcriptional regulatory mechanisms controlling the expression of these molecules. The specific mechanism still needed verification using subsequent mechanism experiments. This may require further extraction of hepatic nuclear proteins and multi-omics testing. In the last revision, we have removed the results concerning FASN, ACLY, and ELOVL6 to avoid introducing conflicting findings. In this current discussion, we have included a section addressing this issue. We emphasize that PKLR and SCD1, along with other lipogenic molecules, are not exclusively regulated by ChREBP. The precise regulatory mechanisms governing these molecules during pregnancy remain incompletely understood and warrant further investigation (*Line 364-374 of Page 17 in manuscript*).

.....
Comment 3: The authors have performed new experiments in Figure 2 with larger numbers

of mice per group to assess the impact of liver ChREBP on insulin resistance of pregnancy in both normal dietary conditions and with high-fructose feeding. The effects of liver ChREBP KO to reduce insulin resistance in the fructose fed state are robust. The original experiment showed minimal or no effect of liver ChREBP KO in on glycemic indices in the chow fed condition. The new experiments do show a statistically significant effect of ChREBP KO to reduce enhance some, but not all, indices of insulin sensitivity though the effect sizes are small. The authors acknowledge that under normal dietary conditions, the influence of hepatic ChREBP on insulin resistance of pregnancy is weak. The authors do not address the second part of this comment. The (new) modest improvements in insulin sensitivity with liver ChREBP KO in normal dietary conditions do seem to parallel a modest reduction in progesterone with liver ChREBP KO (Fig 3C) suggesting that in normal dietary conditions, ChREBP has only a modest effect on progesterone production and the insulin resistance of pregnancy. Again, overall, these results indicate that in normal dietary conditions, liver ChREBP has marginal impact on these important aspects of pregnancy. This should be clear in the abstract and throughout the rest of the manuscript.

Response 3: Thank you for this recommendation. We acknowledge your point that under normal dietary conditions, hepatic ChREBP has modest effect on progesterone production and insulin resistance during pregnancy. In the revised version, we emphasize this in the abstract (*Line 33-37 of Page 2 in manuscript*). In the corresponding results section, we provide detailed descriptions of the effects of liver-specific ChREBP deficiency on insulin resistance and progesterone levels under normal dietary conditions (*Line 141-150 of Page 7, Line 154-155, 172-176 of Page 8 and Line 183-185 of Page 9 in manuscript*). Additionally, we have revised the conclusions in the discussion section, as well as the description of the schematic diagram illustrating the mechanism (*Line 339-342 of Page 16, Line 387-389 of Page 18, Line 463-467 of Page 22 and Line 953-967 of Page 38 in manuscript*). Furthermore, we highlighted that the impact of ChREBP/progesterone on insulin resistance needs to be activated under conditions of high-fructose diet.

.....
Comment 4: The authors propose that fructose feeding induced hepatic ChREBP/MTTP which increases circulating cholesterol and that this drives increased progesterone synthesis exacerbating pregnancy associated insulin resistance. The authors provide new and convincing evidence that fructose feeding can induce progesterone and that this induction is ChREBP dependent (New Figure 3C). This is interesting and exciting. However, the mechanism by which this occurs remains speculative. The authors propose that the ability of fructose to activate ChREBP and induce MTTP enhances cholesterol efflux and that this drives progesterone synthesis by providing more cholesterol substrate for its synthesis. However, there is extensive literature examining biosynthesis of sex hormones and it does not support the idea that cholesterol delivery is limiting for sex steroid synthesis or that increasing circulating cholesterol can drive an increase sex steroid hormone synthesis. As one example, many clinical studies show that statin used which has a large effect to reduce circulating cholesterol in people, has no impact on circulating testosterone in men or estradiol in women. This is likely because the amount of circulating cholesterol FAR exceeds the amount required for sex steroid hormone synthesis by many orders of magnitude. The authors provide evidence that circulating levels of cholesterol do increase with fructose feeding and decline

with ChREBP KO, but there is abundant circulating cholesterol in all conditions (New Fig 4B) and again, this is in far excess of the amount of cholesterol that hormone synthetic tissues require to synthesize sex steroid hormones. The authors do provide very interesting data that MTTP overexpression in liver ChREBP KO mice does increase circulating progesterone. However, it is more likely through some indirect mechanism mediated by lipid droplets or other effects on the liver than by increasing the delivery of cholesterol to be used as substrate. The authors could test their cholesterol delivery hypothesis directly or markedly revise the text around these claims.

Response 4: Thank you for pointing out that cholesterol delivery may not be limiting for sex steroid synthesis, or that increasing circulating cholesterol may not necessarily drive the synthesis of steroid hormones. In the revised version, we have retitled this section of results and provided a revised description of the result figures, removing the previous description of MTTP regulating progesterone synthesis by promoting hepatic cholesterol efflux (*Line 204-205 of Page 10 in manuscript*). Additionally, in the discussion section, we emphasized that the specific mechanism by which MTTP increases progesterone levels remains unclear and proposed several possible mechanisms of action (*Line 437-451 of Page 20 in manuscript*). Furthermore, we have made modifications to the abstract (*Line 28-30 of Page 2 in manuscript*), schematic diagram and its description (*Line 953-967 of Page 38 in manuscript*).

.....
Comment 7: The authors provided data regarding their gene targeting strategy to generate the ChREBP floxed mice used in this manuscript. This is a new model not previously published. Details of the strategy need to be included in the manuscript including targeted exons, at least in supplementary data. The authors have target exon 2 which potentially leaves the alternative translation start site documented to exist in exon 4 intact. The authors should perform additional validation studies to show that a wide range of canonical ChREBP transcriptional targets are downregulated at the mRNA level the in liver ChREBP KO. This should also be included in supplementary data.

Response 7: Thank you for this recommendation. In the revised version, we have added the description of the gene targeting strategy to the methods section (*Line 499-511 of Page 23 in manuscript*). Additionally, representative images identifying the mouse genotypes have been included in the supplementary data (*Fig. S3A*). Furthermore, we conducted RT-qPCR monitoring, and the results indicate that the mRNA levels of ChREBP transcription targets are downregulated in the liver of ChREBP KO mice (*Fig. S3E*).

Dear Prof. Duan,

Thank you for submitting your revised manuscript. I have now looked at everything and all is fine. Therefore, I am very pleased to accept your manuscript for publication in EMBO Reports.

Congratulations on a nice work!

Kind regards,

Deniz Senyilmaz Tiebe

--

Deniz Senyilmaz Tiebe, PhD

Editor

EMBO Reports

--
